# CSOT: Curriculum and Structure-Aware Optimal Transport for Learning with Noisy Labels

**Wanxing Chang**[1]    **Ye Shi**[1,2]    **Jingya Wang**[1,2*]

[1]ShanghaiTech University
[2]Shanghai Engineering Research Center of Intelligent Vision and Imaging

{changwx,shiye,wangjingya}@shanghaitech.edu.cn

## Abstract

Learning with noisy labels (LNL) poses a significant challenge in training a well-generalized model while avoiding overfitting to corrupted labels. Recent advances have achieved impressive performance by identifying clean labels and correcting corrupted labels for training. However, the current approaches rely heavily on the models predictions and evaluate each sample independently without considering either the global or local structure of the sample distribution. These limitations typically result in a suboptimal solution for the identification and correction processes, which eventually leads to models overfitting to incorrect labels. In this paper, we propose a novel optimal transport (OT) formulation, called Curriculum and Structure-aware Optimal Transport (CSOT). CSOT concurrently considers the inter- and intra-distribution structure of the samples to construct a robust denoising and relabeling allocator. During the training process, the allocator incrementally assigns reliable labels to a fraction of the samples with the highest confidence. These labels have both global discriminability and local coherence. Notably, CSOT is a new OT formulation with a nonconvex objective function and curriculum constraints, so it is not directly compatible with classical OT solvers. Here, we develop a lightspeed computational method that involves a scaling iteration within a generalized conditional gradient framework to solve CSOT efficiently. Extensive experiments demonstrate the superiority of our method over the current state-of-the-arts in LNL. Code is available at https://github.com/changwxx/CSOT-for-LNL.

## 1   Introduction

Deep neural networks (DNNs) have significantly boosted performance in various computer vision tasks, including image classification [33], object detection [61], and semantic segmentation [32]. However, the remarkable performance of deep learning algorithms heavily relies on large-scale high-quality human annotations, which are extremely expensive and time-consuming to obtain. Alternatively, mining large-scale labeled data based on a web search and user tags [49, 37] can provide a cost-effective way to collect labels, but this approach inevitably introduces noisy labels. Since DNNs can so easily overfit to noisy labels [4, 79], such label noise can significantly degrade performance, giving rise to a challenging task: learning with noisy labels (LNL) [50, 52, 46].

Numerous strategies have been proposed to mitigate the negative impact of noisy labels, including loss correction based on transition matrix estimation [35], re-weighting [60], label correction [76]

---

*Corresponding author.

37th Conference on Neural Information Processing Systems (NeurIPS 2023).

and sample selection [52]. Recent advances have achieved impressive performance by identifying clean labels and correcting corrupted labels for training. However, current approaches rely heavily on the models predictions to identify or correct labels even if the model is not yet sufficiently trained. Moreover, these approaches often evaluate each sample independently, disregarding the global or local structure of the sample distribution. Hence, the identification and correction process results in a suboptimal solution which eventually leads to a model overfitting to incorrect labels.

In light of the limitations of distribution modeling, optimal transport (OT) offers a promising solution by optimizing the global distribution matching problem that searches for an efficient transport plan from one distribution to another. To date, OT has been applied in various machine learning tasks [11, 83, 28]. In particular, OT-based pseudo-labeling [11, 73] attempts to map samples to class centroids, while considering the *inter-distribution* matching of samples and classes. However, such an approach could also produce assignments that overlook the inherent coherence structure of the sample distribution, *i.e. intra-distribution* coherence. More specifically, the cost matrix in OT relies on pairwise metrics, so two nearby samples could be mapped to two far-away class centroids (Fig. 1).

In this paper, to enhance intra-distribution coherence, we propose a new OT formulation for denoising and relabeling, called Structure-aware Optimal Transport (SOT). This formulation fully considers the intra-distribution structure of the samples and produces robust assignments with both *global discriminability* and *local coherence*. Technically speaking, we introduce local coherent regularized terms to encourage both prediction- and label-level local consistency in the assignments. Furthermore, to avoid generating incorrect labels in the early stages of training or cases with high noise ratios, we devise Curriculum and Structure-aware Optimal Transport (CSOT) based on SOT. CSOT constructs a robust denoising and relabeling allocator by relaxing one of the equality constraints to allow only a fraction of the samples with the highest confidence to be selected. These samples are then assigned with reliable pseudo labels. The allocator progressively selects and relabels batches of high-confidence samples based on an increasing budget factor that controls the number of selected samples. Notably, CSOT is a new OT formulation with a nonconvex objective function and curriculum constraints, so it is significantly different from the classical OT formulations. Hence, to solve CSOT efficiently, we developed a lightspeed computational method that involves a scaling iteration within a generalized conditional gradient framework [59].

Our contribution can be summarized as follows: 1) We tackle the denoising and relabeling problem in LNL from a new perspective, i.e. simultaneously considering the *inter-* and *intra-distribution* structure for generating superior pseudo labels using optimal transport. 2) To fully consider the intrinsic coherence structure of sample distribution, we propose a novel optimal transport formulation, namely Curriculum and Structure-aware Optimal Transport (CSOT), which constructs a robust denoising and relabeling allocator that mitigates error accumulation. This allocator selects a fraction of high-confidence samples, which are then assigned reliable labels with both *global discriminability* and *local coherence*. 3) We further develop a lightspeed computational method that involves a scaling iteration within a generalized conditional gradient framework to efficiently solve CSOT. 4) Extensive experiments demonstrate the superiority of our method over state-of-the-art methods in LNL.

## 2 Related Work

**Learning with noisy labels.** LNL is a well-studied field with numerous strategies having been proposed to solve this challenging problem, such as robust loss design [82, 70], loss correction [35, 56], loss re-weighting [60, 80] and sample selection [52, 31, 41]. Currently, the methods that are delivering superior performance mainly involve learning from both selected clean labels and relabeled corrupted labels [46, 45]. The mainstream approaches for identifying clean labels typically rely on the small-loss criterion [31, 77, 71, 14]. These methods often model per-sample loss distributions using a Beta Mixture Model [51] or a Gaussian Mixture Model [57], treating samples with smaller loss as clean ones [3, 71, 46]. The label correction methods, such as PENCIL [76], Selfie [63], ELR [50], and DivideMix [46], typically adopt a pseudo-labeling strategy that leverages the DNNs predictions to correct the labels. However, these approaches evaluate each sample independently without considering the correlations among samples, which leads to a suboptimal identification and correction solution. To this end, some work [55, 45] attempt to leverage $k$-nearest neighbor predictions [6] for clean identification and label correction. Besides, to further select and correct noisy labels robustly,

OT Cleaner [73], as well as concurrent OT-Filter [23], designed to consider the global sample distribution by formulating pseudo-labeling as an optimal transport problem. In this paper, we propose CSOT to construct a robust denoising and relabeling allocator that simultaneously considers both the global and local structure of sample distribution so as to generate better pseudo labels.

**Optimal transport-based pseudo-labeling.**   OT is a constrained optimization problem that aims to find the optimal coupling matrix to map one probability distribution to another while minimizing the total cost [40]. OT has been formulated as a pseudo-labeling (PL) technique for a range of machine learning tasks, including class-imbalanced learning [44, 28, 68], semi-supervised learning [65, 54, 44], clustering [5, 11, 25], domain adaptation [83, 12], label refinery [83, 68, 73, 23], and others. Unlike prediction-based PL [62], OT-based PL optimizes the mapping samples to class centroids, while considering the global structure of the sample distribution in terms of marginal constraints instead of per-sample predictions. For example, Self-labelling [5] and SwAV [11], which are designed for self-supervised learning, both seek an optimal equal-partition clustering to avoid the models collapse. In addition, because OT-based PL considers marginal constraints, it can also consider class distribution to solve class-imbalance problems [44, 28, 68]. However, these approaches only consider the inter-distribution matching of samples and classes but do not consider the intra-distribution coherence structure of samples. By contrast, our proposed CSOT considers both the inter- and intra-distribution structure and generates superior pseudo labels for noise-robust learning.

**Curriculum learning.**   Curriculum learning (CL) attempts to gradually increase the difficulty of the training samples, allowing the model to learn progressively from easier concepts to more complex ones [42]. CL has been applied to various machine learning tasks, including image classification [38, 84], and reinforcement learning [53, 2]. Recently, the combination of curriculum learning and pseudo-labeling has become popular in semi-supervised learning. These methods mainly focus on dynamic confident thresholding [69, 29, 75] instead of adopting a fixed threshold [62]. Flex-match [78] designs class-wise thresholds and lowers the thresholds for classes that are more difficult to learn. Different from dynamic thresholding approaches, SLA [65] only assigns pseudo labels to easy samples gradually based on an OT-like problem. In the context of LNL, CurriculumNet [30] designs a curriculum by ranking the complexity of the data using its distribution density in a feature space. Alternatively, RoCL [85] selects easier samples considering both the dynamics of the per-sample loss and the output consistency. Our proposed CSOT constructs a robust denoising and relabeling allocator that gradually assigns high-quality labels to a fraction of the samples with the highest confidence. This encourages both global discriminability and local coherence in assignments.

# 3   Preliminaries

**Optimal transport.**   Here we briefly recap the well-known formulation of OT. Given two probability simplex vectors $\boldsymbol{\alpha}$ and $\boldsymbol{\beta}$ indicating two distributions, as well as a cost matrix $\mathbf{C} \in \mathbb{R}^{|\boldsymbol{\alpha}| \times |\boldsymbol{\beta}|}$, where $|\boldsymbol{\alpha}|$ denotes the dimension of $\boldsymbol{\alpha}$, OT aims to seek the optimal coupling matrix $\mathbf{Q}$ by minimizing the following objective

$$\min_{\mathbf{Q} \in \mathbf{\Pi}(\boldsymbol{\alpha}, \boldsymbol{\beta})} \langle \mathbf{C}, \mathbf{Q} \rangle, \tag{1}$$

where $\langle \cdot, \cdot \rangle$ denote Frobenius dot-product. The coupling matrix $\mathbf{Q}$ satisfies the polytope $\mathbf{\Pi}(\boldsymbol{\alpha}, \boldsymbol{\beta}) = \left\{ \mathbf{Q} \in \mathbb{R}_+^{|\boldsymbol{\alpha}| \times |\boldsymbol{\beta}|} | \mathbf{Q} \mathbb{1}_{|\boldsymbol{\beta}|} = \boldsymbol{\alpha}, \ \mathbf{Q}^\top \mathbb{1}_{|\boldsymbol{\alpha}|} = \boldsymbol{\beta} \right\}$, where $\boldsymbol{\alpha}$ and $\boldsymbol{\beta}$ are essentially marginal probability vectors. Intuitively speaking, these two marginal probability vectors can be interpreted as coupling budgets, which control the mapping intensity of each row and column in $\mathbf{Q}$.

**Pseudo-labeling based on optimal transport.**   Let $\mathbf{P} \in \mathbb{R}_+^{B \times C}$ denote classifier softmax predictions, where $B$ is the batch size of samples, and $C$ is the number of classes. The OT-based PL considers mapping samples to class centroids and the cost matrix $\mathbf{C}$ can be formulated as $-\log \mathbf{P}$ [65, 68]. We can rewrite the objective for OT-based PL based on Problem (1) as follows

$$\min_{\mathbf{Q} \in \mathbf{\Pi}(\frac{1}{B} \mathbb{1}_B, \frac{1}{C} \mathbb{1}_C)} \langle -\log \mathbf{P}, \mathbf{Q} \rangle, \tag{2}$$

where $\mathbb{1}_d$ indicates a $d$-dimensional vector of ones. The pseudo-labeling matrix can be obtained by normalization: $B\mathbf{Q}$. Unlike prediction-based PL [62] which evaluates each sample independently,

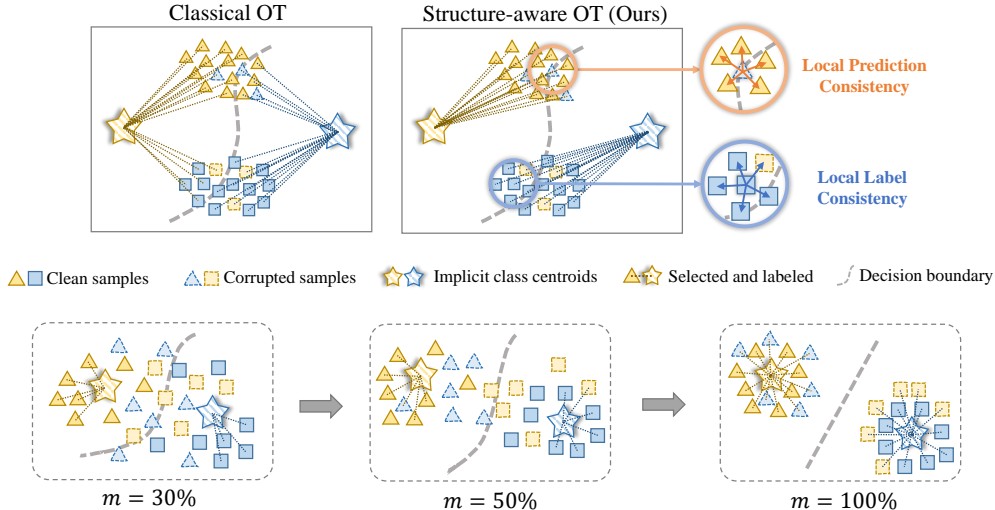

Figure 1: **(Top) Comparison between classical OT and our proposed Structure-aware OT.** Classical OT tends to mismatch two nearby samples to two far-away class centroids when the decision boundary is not accurate enough. To mitigate this, our SOT generates local consensus assignments for each sample by preserving prediction-level and label-level consistency. Notably, for vague samples located near the ambiguous decision boundary, SOT rectifies their assignments based on the neighborhood majority consistency. **(Bottom) The illustration of our curriculum denoising and relabeling based on proposed CSOT.** The decision boundary refers to the surface that separates two classes by the classifier. The $m$ represents the curriculum budget that controls the number of selected samples and progressively increases during the training process.

OT-based PL considers inter-distribution matching of samples and classes, as well as the global structure of sample distribution, thanks to the equality constraints.

**Sinkhorn algorithm for classical optimal transport problem.** Directly optimizing the exact OT problem would be time-consuming, and an entropic regularization term is introduced [19]: $\min_{\mathbf{Q} \in \mathbf{\Pi}(\boldsymbol{\alpha}, \boldsymbol{\beta})} \langle \mathbf{C}, \mathbf{Q} \rangle + \varepsilon \langle \mathbf{Q}, \log \mathbf{Q} \rangle$, where $\varepsilon > 0$. The entropic regularization term enables OT to be approximated efficiently by the Sinkhorn algorithm [19], which involves matrix scaling iterations executed efficiently by matrix multiplication on GPU.

## 4 Methodology

**Problem setup.** Let $\mathcal{D}_{train} = \{(\mathbf{x}_i, y_i)\}_{i=1}^N$ denote the noisy training set, where $\mathbf{x}_i$ is an image with its associated label $y_i$ over $C$ classes, but whether the given label is accurate or not is unknown. We call the correctly-labeled ones as *clean*, and the mislabeled ones as *corrupted*. LNL aims to train a network that is robust to corrupted labels and achieves high accuracy on a clean test set.

### 4.1 Structure-Aware Optimal Transport for Denoising and Relabeling

Even though existing OT-based PL considers the global structure of sample distribution, the intrinsic coherence structure of the samples is ignored. Specifically, the cost matrix in OT relies on pairwise metrics and thus two nearby samples could be mapped to two far-away class centroids. To further consider the intrinsic coherence structure, we propose a Structure-aware Optimal Transport (SOT) for denoising and relabeling, which promotes local consensus assignment by encouraging prediction-level and label-level consistency, as shown in Fig. 1.

Our proposed SOT for denoising and relabeling is formulated by adding two local coherent regularized terms based on Problem (2). Given a cosine similarity $\mathbf{S} \in \mathbb{R}^{B \times B}$ among samples in feature space, a one-hot label matrix $\mathbf{L} \in \mathbb{R}^{B \times C}$ transformed from given noisy labels, and a softmax prediction matrix $\mathbf{P} \in \mathbb{R}_+^{B \times C}$, SOT is formulated as follows

$$\min_{\mathbf{Q} \in \mathbf{\Pi}(\frac{1}{B}\mathbb{1}_B, \frac{1}{C}\mathbb{1}_C)} \langle -\log \mathbf{P}, \mathbf{Q} \rangle + \kappa \left( \Omega^{\mathbf{P}}(\mathbf{Q}) + \Omega^{\mathbf{L}}(\mathbf{Q}) \right), \tag{3}$$

where the local coherent regularized terms $\Omega^{\mathbf{P}}$ and $\Omega^{\mathbf{L}}$ encourages prediction-level and label-level local consistency respectively, and are defined as follows

$$\Omega^{\mathbf{P}}(\mathbf{Q}) = -\sum_{i,j} \mathbf{S}_{ij} \sum_k \mathbf{P}_{ik} \mathbf{P}_{jk} \mathbf{Q}_{ik} \mathbf{Q}_{jk} = -\left\langle \mathbf{S}, (\mathbf{P} \odot \mathbf{Q})(\mathbf{P} \odot \mathbf{Q})^\top \right\rangle, \quad (4)$$

$$\Omega^{\mathbf{L}}(\mathbf{Q}) = -\sum_{i,j} \mathbf{S}_{ij} \sum_k \mathbf{L}_{ik} \mathbf{L}_{jk} \mathbf{Q}_{ik} \mathbf{Q}_{jk} = -\left\langle \mathbf{S}, (\mathbf{L} \odot \mathbf{Q})(\mathbf{L} \odot \mathbf{Q})^\top \right\rangle, \quad (5)$$

where $\odot$ indicates element-wise multiplication. To be more specific, $\Omega^{\mathbf{P}}$ encourages assigning larger weight to $\mathbf{Q}_{ik}$ and $\mathbf{Q}_{jk}$ if the $i$-th sample is very close to the $j$-th sample, and their predictions $\mathbf{P}_{ik}$ and $\mathbf{P}_{jk}$ from the $k$-th class centroid are simultaneously high. Analogously, $\Omega^{\mathbf{L}}$ encourages assigning larger weight to those samples whose neighborhood label consistency is rather high. Unlike the formulation proposed in [1, 16], which focuses on sample-to-sample mapping, our method introduces a sample-to-class mapping that leverages the intrinsic coherence structure within the samples.

## 4.2 Curriculum and Structure-Aware Optimal Transport for Denoising and Relabeling

In the early stages of training or in scenarios with a high noise ratio, the predictions and feature representation would be vague and thus lead to the wrong assignments for SOT. For the purpose of robust clean label identification and corrupted label correction, we further propose a Curriculum and Structure-aware Optimal Transport (CSOT), which constructs a robust curriculum allocator. This curriculum allocator gradually selects a fraction of the samples with high confidence from the noisy training set, controlled by a budget factor, then assigns reliable pseudo labels for them.

Our proposed CSOT for denoising and relabeling is formulated by introducing new curriculum constraints based on SOT in Problem (3). Given curriculum budget factor $m \in [0, 1]$, our CSOT seeks optimal coupling matrix $\mathbf{Q}$ by minimizing following objective

$$\min_{\mathbf{Q}} \langle -\log \mathbf{P}, \mathbf{Q} \rangle + \kappa \left( \Omega^{\mathbf{P}}(\mathbf{Q}) + \Omega^{\mathbf{L}}(\mathbf{Q}) \right)$$

$$\text{s.t.} \quad \mathbf{Q} \in \left\{ \mathbf{Q} \in \mathbb{R}_+^{B \times C} | \mathbf{Q}\mathbb{1}_C \leq \frac{1}{B}\mathbb{1}_B, \mathbf{Q}^\top \mathbb{1}_B = \frac{m}{C}\mathbb{1}_C \right\}. \quad (6)$$

Unlike SOT, which enforces an equality constraint on the samples, CSOT relaxes this constraint and defines the total coupling budget as $m \in [0, 1]$, where $m$ represents the expected total sum of $\mathbf{Q}$. Intuitively speaking, $m = 0.5$ indicates that top $50\%$ confident samples are selected from all the classes, avoiding only selecting the same class for all the samples within a mini-batch. And the budget $m$ progressively increases during the training process, as shown in Fig. 1.

Based on the optimal coupling matrix $\mathbf{Q}$ solved from Problem (6), we can obtain pseudo label by argmax operation, *i.e.* $\hat{y}_i = \arg\max_j \mathbf{Q}_{ij}$. In addition, we define the general confident scores of samples as $\mathcal{W} = \{w_0, w_1, \cdots, w_{B-1}\}$, where $w_i = \mathbf{Q}_{i\hat{y}_i}/(m/C)$. Since our curriculum allocator assigns weight to only a fraction of samples controlled by $m$, we use $\text{topK}(\mathcal{S}, k)$ operation (return top-$k$ indices of input set $\mathcal{S}$) to identify selected samples denoted as $\delta_i$

$$\delta_i = \begin{cases} 1, & i \in \text{topK}(\mathcal{W}, \lfloor mB \rfloor) \\ 0, & \text{otherwise} \end{cases}, \quad (7)$$

where $\lfloor \cdot \rfloor$ indicates the round down operator. Then the noisy dataset $\mathcal{D}_{train}$ can be splited into $\mathcal{D}_{clean}$ and $\mathcal{D}_{corrupted}$ as follows

$$\mathcal{D}_{clean} \leftarrow \{(\mathbf{x}_i, y_i, w_i)|\hat{y}_i = y_i, \delta_i = 1, (\mathbf{x}_i, y_i) \in \mathcal{D}_{train}\},$$
$$\mathcal{D}_{corrupted} \leftarrow \{(\mathbf{x}_i, \hat{y}_i, w_i)|\hat{y}_i \neq y_i, (\mathbf{x}_i, y_i) \in \mathcal{D}_{train}\}. \quad (8)$$

## 4.3 Training Objectives

To avoid error accumulation in the early stage of training, we adopt a two-stage training scheme. In the first stage, the model is supervised by progressively selected clean labels and self-supervised by unselected samples. In the second stage, the model is semi-supervised by all denoised labels. Notably, we construct our training objective mainly based on Mixup loss $\mathcal{L}^{mix}$ and Label consistency

loss $\mathcal{L}^{lab}$ same as NCE [45], and a self-supervised loss $\mathcal{L}^{simsiam}$ proposed in SimSiam [15]. The detailed formulations of mentioned loss and training process are given in Appendix. Our two-stage training objective can be constructed as follows

$$\mathcal{L}^{sup} = \mathcal{L}^{mix}_{\mathcal{D}_{clean}} + \mathcal{L}^{lab}_{\mathcal{D}_{clean}} + \lambda_1 \mathcal{L}^{simsiam}_{\mathcal{D}_{corrupted}}, \tag{9}$$

$$\mathcal{L}^{semi} = \mathcal{L}^{mix}_{\mathcal{D}_{clean}} + \mathcal{L}^{lab}_{\mathcal{D}_{clean}} + \lambda_2 \mathcal{L}^{lab}_{\mathcal{D}_{corrupted}}. \tag{10}$$

## 5 Lightspeed Computation for CSOT

The proposed CSOT is a new OT formulation with nonconvex objective function and curriculum constraints, which cannot be solved directly by classical OT solvers. To this end, we develop a lightspeed computational method that involves a scaling iteration within a generalized conditional gradient framework to solve CSOT efficiently. Specifically, we first introduce an efficient scaling iteration for solving the OT problem with curriculum constraints without considering the local coherent regularized terms, *i.e.* Curriculum OT (COT). Then, we extend our approach to solve the proposed CSOT problem, which involves a nonconvex objective function and curriculum constraints.

### 5.1 Solving Curriculum Optimal Transport

For convenience, we formulate curriculum constraints in Probelm (6) in a more general form. Given two vectors $\boldsymbol{\alpha}$ and $\boldsymbol{\beta}$ that satisfy $\|\boldsymbol{\alpha}\|_1 \geq \|\boldsymbol{\beta}\|_1 = m$, a general polytope of curriculum constraints $\boldsymbol{\Pi}^c(\boldsymbol{\alpha}, \boldsymbol{\beta})$ is formulated as

$$\boldsymbol{\Pi}^c(\boldsymbol{\alpha}, \boldsymbol{\beta}) = \left\{ \mathbf{Q} \in \mathbb{R}_+^{|\boldsymbol{\alpha}| \times |\boldsymbol{\beta}|} | \mathbf{Q} \mathbb{1}_{|\boldsymbol{\beta}|} \leq \boldsymbol{\alpha}, \mathbf{Q}^\top \mathbb{1}_{|\boldsymbol{\alpha}|} = \boldsymbol{\beta} \right\}. \tag{11}$$

For the efficient computation purpose, we consider an entropic regularized version of COT

$$\min_{\mathbf{Q} \in \boldsymbol{\Pi}^c(\boldsymbol{\alpha}, \boldsymbol{\beta})} \langle \mathbf{C}, \mathbf{Q} \rangle + \varepsilon \langle \mathbf{Q}, \log \mathbf{Q} \rangle, \tag{12}$$

where we denote the cost matrix $\mathbf{C} := -\log \mathbf{P}$ in Probelm (6) for simplicity. Inspired by [8], Problem (12) can be easily re-written as the Kullback-Leibler (KL) projection: $\min_{\mathbf{Q} \in \boldsymbol{\Pi}^c(\boldsymbol{\alpha}, \boldsymbol{\beta})} \varepsilon \text{KL}(\mathbf{Q} | e^{-\mathbf{C}/\varepsilon})$. Besides, the polytope $\boldsymbol{\Pi}^c(\boldsymbol{\alpha}, \boldsymbol{\beta})$ can be expressed as an intersection of two convex but not affine sets, *i.e.*

$$\mathcal{C}_1 \stackrel{\text{def}}{=} \left\{ \mathbf{Q} \in \mathbb{R}_+^{|\boldsymbol{\alpha}| \times |\boldsymbol{\beta}|} | \mathbf{Q} \mathbb{1}_{|\boldsymbol{\beta}|} \leq \boldsymbol{\alpha} \right\} \quad \text{and} \quad \mathcal{C}_2 \stackrel{\text{def}}{=} \left\{ \mathbf{Q} \in \mathbb{R}_+^{|\boldsymbol{\alpha}| \times |\boldsymbol{\beta}|} | \mathbf{Q}^\top \mathbb{1}_{|\boldsymbol{\alpha}|} = \boldsymbol{\beta} \right\}. \tag{13}$$

In light of this, Problem (12) can be solved by performing iterative KL projection between $\mathcal{C}_1$ and $\mathcal{C}_2$, namely Dykstra's algorithm [21] shown in Appendix.

**Lemma 1.** *(Efficient scaling iteration for Curriculum OT) When solving Problem (12) by iterating Dykstra's algorithm, the matrix $\mathbf{Q}^{(n)}$ at $n$ iteration is a diagonal scaling of $\mathbf{K} := e^{-\mathbf{C}/\varepsilon}$, which is the element-wise exponential matrix of $-\mathbf{C}/\varepsilon$:*

$$\mathbf{Q}^{(n)} = \text{diag}\left(\boldsymbol{u}^{(n)}\right) \mathbf{K} \text{diag}\left(\boldsymbol{v}^{(n)}\right), \tag{14}$$

*where the vectors $\boldsymbol{u}^{(n)} \in \mathbb{R}^{|\boldsymbol{\alpha}|}$, $\boldsymbol{v}^{(n)} \in \mathbb{R}^{|\boldsymbol{\beta}|}$ satisfy $\boldsymbol{v}^{(0)} = \mathbb{1}_{|\boldsymbol{\beta}|}$ and follow the recursion formula*

$$\boldsymbol{u}^{(n)} = \min\left(\frac{\boldsymbol{\alpha}}{\mathbf{K}\boldsymbol{v}^{(n-1)}}, \mathbb{1}_{|\boldsymbol{\alpha}|}\right) \quad \text{and} \quad \boldsymbol{v}^{(n)} = \frac{\boldsymbol{\beta}}{\mathbf{K}^\top \boldsymbol{u}^{(n)}}. \tag{15}$$

The proof is given in the Appendix. Lemma 1 allows a fast implementation of Dykstra's algorithm by only performing matrix-vector multiplications. This scaling iteration for entropic regularized COT is very similar to the widely-used and efficient Sinkhorn Algorithm [19], as shown in Algorithm 1.

### 5.2 Solving Curriculum and Structure-Aware Optimal Transport

In the following, we propose to solve CSOT within a Generalized Conditional Gradient (GCG) algorithm [59] framework, which strongly relies on computing Curriculum OT by scaling iterations

---

**Algorithm 1** Efficient scaling iteration for entropic regularized Curriculum OT

---

1: **Input:** Cost matrix $\mathbf{C}$, marginal constraints vectors $\boldsymbol{\alpha}$ and $\boldsymbol{\beta}$, entropic regularization weight $\varepsilon$
2: Initialize: $\mathbf{K} \leftarrow e^{-\mathbf{C}/\varepsilon}$, $\boldsymbol{v}^{(0)} \leftarrow \mathbb{1}_{|\boldsymbol{\beta}|}$
3: Compute: $\mathbf{K}_{\boldsymbol{\alpha}} \leftarrow \frac{\mathbf{K}}{\mathrm{diag}(\boldsymbol{\alpha})\mathbb{1}_{|\boldsymbol{\alpha}|\times|\boldsymbol{\beta}|}}$, $\mathbf{K}_{\boldsymbol{\beta}}^{\top} \leftarrow \frac{\mathbf{K}^{\top}}{\mathrm{diag}(\boldsymbol{\beta})\mathbb{1}_{|\boldsymbol{\beta}|\times|\boldsymbol{\alpha}|}}$ // Saving computation
4: **for** $n = 1, 2, 3, \ldots$ **do**
5:     $\boldsymbol{u}^{(n)} \leftarrow \min\left(\frac{\mathbb{1}_{|\boldsymbol{\alpha}|}}{\mathbf{K}_{\boldsymbol{\alpha}}\boldsymbol{v}^{(n-1)}}, \mathbb{1}_{|\boldsymbol{\alpha}|}\right)$
6:     $\boldsymbol{v}^{(n)} \leftarrow \frac{\mathbb{1}_{|\boldsymbol{\beta}|}}{\mathbf{K}_{\boldsymbol{\beta}}^{\top}\boldsymbol{u}^{(n)}}$
7: **end for**
8: **Return:** $\mathrm{diag}(\boldsymbol{u}^{(n)})\mathbf{K}\mathrm{diag}(\boldsymbol{v}^{(n)})$

---

in Algorithm 1. The conditional gradient algorithm [27, 36] has been used for some penalized OT problems [24, 17] or nonconvex Gromov-Wasserstein distances [58, 67, 13], which can be used to solve Problem (3) directly.

For simplicity, we denote the local coherent regularized terms as $\Omega(\cdot) := \Omega^{\mathbf{P}}(\cdot) + \Omega^{\mathbf{L}}(\cdot)$, and give an entropic regularized CSOT formulation as follows:

$$\min_{\mathbf{Q}\in\boldsymbol{\Pi}^c(\boldsymbol{\alpha},\boldsymbol{\beta})} \langle\mathbf{C}, \mathbf{Q}\rangle + \kappa\Omega(\mathbf{Q}) + \varepsilon\langle\mathbf{Q}, \log\mathbf{Q}\rangle. \tag{16}$$

Since the local coherent regularized term $\Omega^{\mathbf{P}}(\cdot)$ is differentiable, Problem (16) can be solved within the GCG algorithm framework, shown in Algorithm 2. And the linearization procedure in Line 5 can be computed efficiently by the scaling iteration proposed in Sec 5.1.

---

**Algorithm 2** Generalized conditional gradient algorithm for entropic regularized CSOT

---

1: **Input:** Cost matrix $\mathbf{C}$, marginal constraints vectors $\boldsymbol{\alpha}$ and $\boldsymbol{\beta}$, entropic regularization weight $\varepsilon$, local coherent regularization weight $\kappa$, local coherent regularization function $\Omega : \mathbb{R}^{|\boldsymbol{\alpha}|\times|\boldsymbol{\beta}|} \rightarrow \mathbb{R}$, and its gradient function $\nabla\Omega : \mathbb{R}^{|\boldsymbol{\alpha}|\times|\boldsymbol{\beta}|} \rightarrow \mathbb{R}^{|\boldsymbol{\alpha}|\times|\boldsymbol{\beta}|}$
2: Initialize: $\mathbf{Q}^{(0)} \leftarrow \boldsymbol{\alpha}\boldsymbol{\beta}^T$
3: **for** $i = 1, 2, 3, \ldots$ **do**
4:     $\mathbf{G}^{(i)} \leftarrow \mathbf{Q}^{(i)} + \kappa\nabla\Omega(\mathbf{Q}^{(i)})$ // Gradient computation
5:     $\widetilde{\mathbf{Q}}^{(i)} \leftarrow \mathrm{argmin}_{\mathbf{Q}\in\boldsymbol{\Pi}^c(\boldsymbol{\alpha},\boldsymbol{\beta})} \langle\mathbf{Q}, \mathbf{G}^{(i)}\rangle + \varepsilon\langle\mathbf{Q}, \log\mathbf{Q}\rangle$
    // Linearization, solved efficiently by Algorithm 1
6:     Choose $\eta^{(i)} \in [0, 1]$ so that it satisfies the Armijo rule // Backtracking line-search
7:     $\mathbf{Q}^{(i+1)} \leftarrow \left(1 - \eta^{(i)}\right)\mathbf{Q}^{(i)} + \eta^{(i)}\widetilde{\mathbf{Q}}^{(i)}$ // Update
8: **end for**
9: **Return:** $\mathbf{Q}^{(i)}$

---

## 6 Experiments

### 6.1 Implementation Details

We conduct experiments on three standard LNL benchmark datasets: CIFAR-10 [43], CIFAR-100 [43] and Webvision [49]. We follow most implementation details from the previous work DivideMix [46] and NCE [45]. Here we provide some specific details of our approach. The warm-up epochs are set to 10/30/10 for CIFAR-10/100/Webvision respectively. For CIFAR-10/100, the supervised learning epoch $T_{sup}$ is set to 250, and the semi-supervised learning epoch $T_{semi}$ is set to 200. For Webvision, $T_{sup} = 80$ and $T_{semi} = 70$. For all experiments, we set $\lambda_1 = 1, \lambda_2 = 1, \varepsilon = 0.1, \kappa = 1$. And we adopt a simple linear ramp for curriculum budget, *i.e.* $m = \min(1.0, m_0 + \frac{t-1}{T_{sup}-1})$ with an initial budget $m_0 = 0.3$. For the GCG algorithm, the number of outer loops is set to 10, and the number for inner scaling iteration is set to 100. The batch size $B$ for denoising and relabeling is set to 1024. More details will be provided in Appendix.

Table 1: **Comparison with state-of-the-art methods in test accuracy (%) on CIFAR-10 and CIFAR-100.** The results are mainly copied from [45, 48]. We present the performance of our CSOT method using the "mean±variance" format, which is obtained from 3 trials with different seeds.

| Dataset | CIFAR-10 | | | | | CIFAR-100 | | | |
| --- | --- | --- | --- | --- | --- | --- | --- | --- | --- |
| Noise type | Symmetric | | | | Assymetric | Symmetric | | | |
| Method/Noise ratio | 0.2 | 0.5 | 0.8 | 0.9 | 0.4 | 0.2 | 0.5 | 0.8 | 0.9 |
| Cross-Entropy | 86.8 | 79.4 | 62.9 | 42.7 | 85.0 | 62.0 | 46.7 | 19.9 | 10.1 |
| F-correction [56] | 86.8 | 79.8 | 63.3 | 42.9 | 87.2 | 61.5 | 46.6 | 19.9 | 10.2 |
| Co-teaching+ [31] | 89.5 | 85.7 | 67.4 | 47.9 | - | 65.6 | 51.8 | 27.9 | 13.7 |
| PENCIL [76] | 92.4 | 89.1 | 77.5 | 58.9 | 88.5 | 69.4 | 57.5 | 31.1 | 15.3 |
| DivideMix [46] | 96.1 | 94.6 | 93.2 | 76.0 | 93.4 | 77.3 | 74.6 | 60.2 | 31.5 |
| ELR [50] | 95.8 | 94.8 | 93.3 | 78.7 | 93.0 | 77.6 | 73.6 | 60.8 | 33.4 |
| NGC [72] | 95.9 | 94.5 | 91.6 | 80.5 | 90.6 | 79.3 | 75.9 | 62.7 | 29.8 |
| RRL [48] | 96.4 | 95.3 | 93.3 | 77.4 | 92.6 | 80.3 | 76.0 | 61.1 | 33.1 |
| MOIT [55] | 93.1 | 90.0 | 79.0 | 69.6 | 92.0 | 73.0 | 64.6 | 46.5 | 36.0 |
| UniCon [41] | 96.0 | 95.6 | 93.9 | **90.8** | 94.1 | 78.9 | 77.6 | 63.9 | 44.8 |
| NCE [45] | 96.2 | 95.3 | 93.9 | 88.4 | 94.5 | **81.4** | 76.3 | 64.7 | 41.1 |
| OT Cleaner [73] | 91.4 | 85.4 | 56.9 | - | - | 67.4 | 58.9 | 31.2 | - |
| OT-Filter [23] | 96.0 | 95.3 | 94.0 | 90.5 | 95.1 | 76.7 | 73.8 | 61.8 | 42.8 |
| **CSOT (Best)** | **96.6±0.10** | **96.2±0.11** | **94.4±0.16** | 90.7±0.33 | **95.5±0.06** | 80.5±0.28 | **77.9±0.18** | **67.8±0.23** | **50.5±0.46** |
| **CSOT (Last)** | 96.4±0.18 | 96.0±0.11 | 94.3±0.20 | 90.5±0.36 | 95.2±0.12 | 80.2±0.31 | 77.7±0.14 | 67.6±0.36 | 50.3±0.33 |

Table 2: **Comparison with SOTA methods in top-1 / 5 test accuracy (%) on the Webvision and ImageNet ILSVRC12 validation sets.**

| | Webvision | | ILSVRC12 | |
| --- | --- | --- | --- | --- |
| Method | top-1 | top-5 | top-1 | top-5 |
| F-correction [56] | 61.12 | 82.68 | 57.36 | 82.36 |
| Decoupling [52] | 62.54 | 84.74 | 58.26 | 82.26 |
| MentorNet [39] | 63.00 | 81.40 | 57.80 | 79.92 |
| Co-teaching [31] | 63.58 | 85.20 | 61.48 | 84.70 |
| DivideMix [46] | 77.32 | 91.64 | 75.20 | 90.84 |
| ELR [50] | 76.26 | 91.26 | 68.71 | 87.84 |
| ELR+ [50] | 77.78 | 91.68 | 70.29 | 89.76 |
| NGC [72] | 79.20 | 91.80 | 74.40 | 91.00 |
| RRL [48] | 77.80 | 91.30 | 74.40 | 90.90 |
| UniCon [41] | 77.60 | 93.44 | 75.29 | 93.72 |
| MOIT [55] | 77.90 | 91.90 | 73.80 | 91.70 |
| NCE [45] | 79.50 | **93.80** | 76.30 | **94.10** |
| **CSOT** | **79.67** | 91.95 | **76.64** | 91.67 |

Table 3: **Time cost (s) for solving CSOT optimization problem of different input sizes.** VDA indicates vanilla Dykstras algorithm-based CSOT solver, while ESI indicates the efficient scaling iteration-based solver.

| $(|\alpha|, |\beta|)$ | VDA-based | ESI-based (Ours) |
| --- | --- | --- |
| (1024,10) | 0.83 | **0.82** ↓ |
| (1024,50) | 1.00 | **0.80** ↓ |
| (1024,100) | 0.87 | **0.80** ↓ |
| (50,50) | 0.82 | **0.79** ↓ |
| (100,100) | 0.88 | **0.80** ↓ |
| (500,500) | 0.88 | **0.87** ↓ |
| (1000,1000) | 0.94 | **0.81** ↓ |
| (2000,2000) | 2.11 | **0.98** ↓ |
| **(3000,3000)** | 3.74 | **0.99** ↓ |

## 6.2 Comparison with the State-of-the-Arts

**Synthetic noisy datasets.** Our method is validated on two synthetic noisy datasets, *i.e.* CIFAR-10 [43] and CIFAR-100 [43]. Following [46, 45], we conduct experiments with two types of label noise: *symmetric* and *asymmetric*. Symmetric noise is injected by randomly selecting a percentage of samples and replacing their labels with random labels. Asymmetric noise is designed to mimic the pattern of real-world label errors, *i.e.* labels are only changed to similar classes (*e.g.* cat↔dog). As shown in Tab. 1, our CSOT has surpassed all the state-of-the-art works across most of the noise ratios. In particular, our CSOT outperforms the previous state-of-the-art method NCE [45] by 2.3%, 3.1% and 9.4% under a high noise rate of CIFAR-10 sym-0.8, CIFAR-100 sym-0.8/0.9, respectively.

**Real-world noisy datasets.** Additionally, we conduct experiments on a large-scale dataset with real-world noisy labels, *i.e.* WebVision [49]. WebVision contains 2.4 million images crawled from the web using the 1,000 concepts in ImageNet ILSVRC12 [20]. Following previous works [46, 45], we conduct experiments only using the first 50 classes of the Google image subset for a total of ∼61,000 images. As shown in Tab. 2, our CSOT surpasses other methods in top-1 accuracy on both Webvision and ILSVRC12 validation sets, demonstrating its superior performance in dealing with real-world noisy datasets. Even though NCE achieves better top-5 accuracy, it suffers from high time costs (using a single NVIDIA A100 GPU) due to the co-training scheme, as shown in Tab. S5.

Table 4: **Ablation studies under multiple label noise ratios on CIFAR-10 and CIFAR-100.** "repl." is an abbreviation for "replaced", and $\mathcal{L}^{ce}$ represents a cross-entropy loss. GMM refers to the selection of clean labels based on small-loss criterion [46]. CT (confidence thresholding [62]) is a relabeling scheme where we set the CT value to 0.95.

| | Dataset | CIFAR-10 | | | | CIFAR-100 | | | |
| | Noise type | | Sym. | | Asym. | | Sym. | | Avg |
| | Method/Noise ratio | 0.5 | 0.8 | 0.9 | 0.4 | 0.5 | 0.8 | 0.9 | |
|---|---|---|---|---|---|---|---|---|---|
| Denoise Relabeling Technique | (a) Classical OT | 95.45 | 91.95 | 82.35 | 95.04 | 75.96 | 62.46 | 43.28 | 78.07 |
| | (b) Structure-aware OT | 95.86 | 91.87 | 83.29 | 95.06 | 76.20 | 63.73 | 44.57 | 78.65 |
| | (c) CSOT w/o $\Omega^P$ and $\Omega^L$ | 95.53 | 93.84 | 89.50 | 95.14 | 75.96 | 66.50 | 47.55 | 80.57 |
| | (d) CSOT w/o $\Omega^P$ | 95.77 | 94.08 | 89.97 | 95.35 | 76.09 | 66.79 | 48.13 | 80.88 |
| | (e) CSOT w/o $\Omega^L$ | 95.55 | 93.97 | 90.41 | 95.15 | 76.17 | 67.28 | 48.01 | 80.93 |
| Learning Technique | (f) GMM + $\mathcal{L}^{sup}$ | 92.48 | 80.37 | 31.76 | 90.80 | 69.52 | 48.49 | 20.86 | 62.04 |
| | (g) CSOT repl. $\mathcal{L}^{sup}$ with $\mathcal{L}^{ce}$ | 93.47 | 81.93 | 53.45 | 91.43 | 72.66 | 50.62 | 21.77 | 66.48 |
| | (h) CSOT w/o $\mathcal{L}^{semi}$ | 95.34 | 93.04 | 88.9 | 94.11 | 75.16 | 61.13 | 36.94 | 77.80 |
| | (i) CSOT repl. correction with CT (0.95) | 95.46 | 90.73 | 89.09 | 95.21 | 75.85 | 64.28 | 48.76 | 79.91 |
| | (j) CSOT w/o $\mathcal{L}^{simsiam}_{\mathcal{D}_{corrupted}}$ | 95.92 | 94.17 | 89.31 | 95.16 | 76.38 | 66.17 | 45.56 | 80.38 |
| | CSOT | **96.20** | **94.39** | **90.65** | **95.50** | **77.94** | **67.78** | **50.50** | **81.85** |

## 6.3 Ablation Studies and Analysis

**Effectiveness of CSOT-based denoising and relabeling.** To verify the effectiveness of each component in our CSOT, we conduct comprehensive ablation experiments, shown in Tab. 4. Compared to classical OT, Structure-aware OT, and Curriculum OT, our proposed CSOT has achieved superior performance. Specifically, our proposed local coherent regularized terms $\Omega^P$ and $\Omega^Q$ indeed contribute to CSOT, as demonstrated in Tab 4 (c)(d)(e). Furthermore, our proposed curriculum constraints yield an improvement of approximately $2\%$ for both classical OT and structure-aware OT, as shown in Tab 4 (a)(b)(c). Particularly, under high noise ratios, the improvement can reach up to $4\%$, which demonstrates the effectiveness of the curriculum relabeling scheme.

**Effectiveness of clean labels identification via CSOT.** As shown in Tab. 4 (f), replacing our CSOT-based denoising and relabeling with GMM [46] for clean label identification significantly degrades the model performance. This phenomenon can be explained by the clean accuracy during training (Fig. 2a) and clean recall rate (Fig. 2c), in which our CSOT consistently outperforms other methods in accurately retrieving clean labels, leading to significant performance improvements. These experiments fully show that our CSOT can maintain both high quantity and high quality of clean labels during training.

**Effectiveness of corrupted labels correction via CSOT.** As shown in Tab. 4 (h), only training with identified clean labels leads to inferior model performance. Furthermore, replacing our CSOT-based denoising and relabeling with confidence thresholding (CT) [62] for corrupted label correction also degrades the model performance, as shown in Tab. 4 (i). The CT methods assign pseudo labels to samples based on model prediction, which is unreliable in the early training stage, especially under high noise rates. Our CSOT-based denoising and relabeling fully consider the inter- and intra-distribution structure of samples, yielding more robust labels. Particularly, our CSOT outperforms NCE and DivideMix significantly in label correction, as demonstrated by the superior corrected accuracy in Fig. 2b and the improved clarity of the confusion matrix in Fig. S7.

**Effectiveness of curriculum training scheme.** According to the progressive clean and corrupted accuracy during the training process shown in Fig. 2a and Fig. 2b, our curriculum identification scheme ensures high accuracy in the early training stage, avoiding overfitting to wrong corrected labels. Note that since our model is trained using only a fraction of clean samples, it is crucial to employ a powerful supervised learning loss to facilitate better learning. Otherwise, the performance will be poor without the utilization of a powerful supervised training loss, as evidenced in Tab. 4 (g). In addition, the incorporation of self-supervised loss enhances noise-robust representation, particularly in high noise rate scenarios, as demonstrated in our experiments in Tab. 4 (j).

**Time cost discussion for solving CSOT** To verify the efficiency of our proposed lightspeed scaling iteration, we conduct some experiments for solving CSOT optimization problem of different input sizes on a single GPU NVIDIA A100. As demonstrated in Tab. 3, our proposed lightspeed

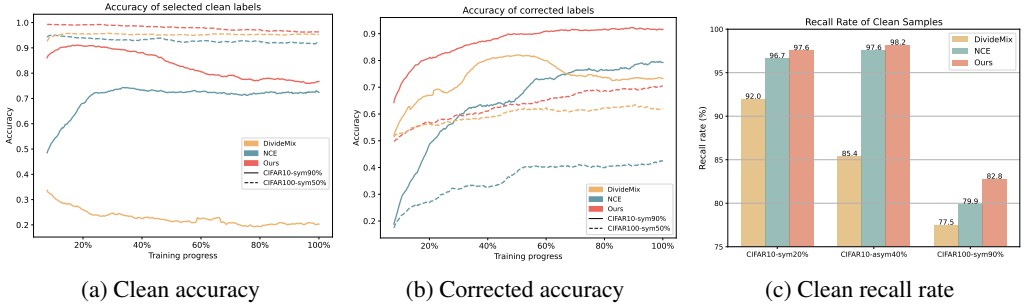

|  (a) Clean accuracy | (b) Corrected accuracy | (c) Clean recall rate |

Figure 2: **Performance comparison for clean label identification and corrupted label correction.**

computational method that involves an efficient scaling iteration (Algorithm 1) achieves lower time cost compared to vanilla Dykstras algorithm (Algorithm S6). Specifically, compared to the vanilla Dykstra-based approach, our efficient scaling iteration version can achieve a speedup of up to 3.7 times, thanks to efficient matrix-vector multiplication instead of matrix-matrix multiplication. Moreover, even for very large input sizes, the computational time cost does not increase significantly.

# 7 Conclusion and Limitation

In this paper, we proposed Curriculum and Structure-aware Optimal Transport (CSOT), a novel solution to construct robust denoising and relabeling allocator that simultaneously considers the inter- and intra-distribution structure of samples. Unlike current approaches, which rely solely on the model's predictions, CSOT considers the global and local structure of the sample distribution to construct a robust denoising and relabeling allocator. During the training process, the allocator assigns reliable labels to a fraction of the samples with high confidence, ensuring both global discriminability and local coherence. To efficiently solve CSOT, we developed a lightspeed computational method that involves a scaling iteration within a generalized conditional gradient framework. Extensive experiments on three benchmark datasets validate the efficacy of our proposed method. While class-imbalance cases are not considered in this paper within the context of LNL, we believe that our approach can be further extended for this purpose.

# 8 Acknowledgement

This work was supported by NSFC (No.62303319), Shanghai Sailing Program (21YF1429400, 22YF1428800), Shanghai Local College Capacity Building Program (23010503100), Shanghai Frontiers Science Center of Human-centered Artificial Intelligence (ShangHAI), MoE Key Laboratory of Intelligent Perception and Human-Machine Collaboration (ShanghaiTech University), and Shanghai Engineering Research Center of Intelligent Vision and Imaging.

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
