# A  Supplement for Training Details

## A.1  Implementation Details

**CIFAR10/100.**   Following previous works [46, 45], we use PreAct ResNet-18 [34] as the backbone, and train it using SGD with a momentum of 0.9, a weight decay of 0.0005, and a batch size of 128. We set the initial learning rate as 0.02, with a cosine learning rate decay schedule. The hidden layer in SimSiam projection MLP is set to 128-d.

**Webvision.**   Following previous works [46, 45], we use inception-resnet v2 [64] as the backbone, and train it using SGD with a momentum of 0.9, a weight decay of 0.0005, and a batch size of 32. We set the initial learning rate as 0.01, with a cosine learning rate decay schedule. The hidden layer in SimSiam projection MLP is set to 384-d.

**Other details.**   All experiments are implemented on a single GPU of NVIDIA A100 with 80 GB memory. We follow DivideMix [46] and NCE [45] to set the hyper-parameters in the mixup loss and label consistency loss. The loss trade-off weights $\lambda_1$ and $\lambda_2$ are empirically set to 1, which is similar to NCE [45]. The selection criterion of the hyper-parameters $\varepsilon$ and $\kappa$ in CSOT formulation is analyzed in Sec. B.5. Our code is modified based on DivideMix [46] https://github.com/LiJunnan1992/DivideMix and NCE [45] https://github.com/lijichang/LNL-NCE. The CSOT solver code is modified based on POT [26].

## A.2  Training Loss

To be self-contained, we specify the Mixup loss $\mathcal{L}^{mix}$ and label consistency loss $\mathcal{L}^{lab}$ adopted in NCE [45], and the self-supervised loss $\mathcal{L}^{simsiam}$ proposed in SimSiam [15].

**Mixup loss.**   Mixup [81] can effectively mitigate noise memorization, and thus mixup regularization can be used to construct augmented samples through linear combinations of existing samples from $\mathcal{D}_{clean}$. Given two existing samples $(\mathbf{x}_i, y_i)$ and $(\mathbf{x}_j, y_j)$ from $\mathcal{D}_{clean}$, an augmented sample $\widetilde{\mathbf{x}}, \widetilde{y}$ can be generated as follows:

$$\widetilde{\mathbf{x}} = \gamma \mathbf{x}_i + (1 - \gamma)\mathbf{x}_j, \quad \widetilde{y} = \gamma p_y(y_i) + (1 - \gamma)p_y(y_j), \tag{S17}$$

where $p_y(y_i)$ is the one-hot vector for the given label $y_i$ and $\gamma \sim Beta(\alpha)$ is a mixup ratio and $\alpha$ is a scalar parameter of Beta distribution. The cross-entropy loss applied to $B'$ augmented samples in each training mini-batch is defined as follows:

$$\mathcal{L}^{mix} = -\frac{1}{B'} \sum_{i=1}^{B'} \widetilde{y}_i \log p(y|\widetilde{\mathbf{x}}_i), \tag{S18}$$

where $p(y|\widetilde{\mathbf{x}}_b)$ is the softmax prediction of a mixup input $\widetilde{\mathbf{x}}_b$.

**Label consistency loss.**   Label consistency regularization encourages the fine-tuned model to produce the same output when there are minor perturbations in the input [62]. Hence consistency regularization can be used to further enhance the robustness of the model [22]. The label consistency is enforced by minimizing the following loss:

$$\mathcal{L}^{lab} = -\frac{1}{B'} \sum_{i=1}^{B'} p_y(y_i) \log p(y|\mathbf{Aug}(x_i)), \tag{S19}$$

where $\mathbf{Aug}(\cdot)$ denotes the function that perturbs the chosen samples using Autoaugment technique proposed in [18].

**SimSiam loss.**   We simply define a feature extractor as $f$ and a projection layer as $h$. Given two augmented views $\mathbf{x}_i^1$ and $\mathbf{x}_i^2$ from an image $\mathbf{x}$, we can have $p_i^1 = h(f(\mathbf{x}_i^1))$ and $z_i^2 = f(\mathbf{x}_i^2)$. The negative cos similarity is defined as follows:

$$\ell(p_i^1, z_i^2) = -\frac{p_i^1 z_i^2}{\|p_i^1\|_2 \|z_i^2\|_2} \tag{S20}$$

where $\|\cdot\|_2$ is $\ell_2$-norm. To construct the contrastive loss by enforcing the consistency between two positive pairs $(p_i^1, z_i^2)$ and $(p_i^2, z_i^1)$, the SimSiam loss is defined as follows:

$$\mathcal{L}^{simsiam} = -\frac{1}{2B'} \sum_{i=1}^{B'} \left( \ell(p_i^1, \texttt{stopgrad}(z_i^2)) + \ell(p_i^2, \texttt{stopgrad}(z_i^1)) \right) \tag{S21}$$

where $\texttt{stopgrad}(\cdot)$ is a stop-gradient operation that can be easily realized by $\texttt{.detach()}$ operation in PyTorch.

### A.3 Training Process

---

**Algorithm S3** Training process of proposed CSOT

---

1: **Input:** Training dataset $\mathcal{D}_{train}$, number of warmup training epochs $T_{warm}$, number of supervised training epochs $T_{sup}$, number of semi-supervised training epochs $T_{semi}$, initial curriculum budget $m_0$.
2: Initialize model parameter $\theta$.
3: **for** $t = 1, \ldots, (T_{sup} + T_{semi})$ **do**
4:     **if** $t < T_{warm}$ **then**
5:         WarmUp($\mathcal{D}_{train}; \theta$).
6:     **else**
7:         Compute the curriculum budget $m = \min(1.0, m_0 + \frac{t-1}{T_{sup}-1})$.
8:         **for** $b = 1, \ldots, N_{batch}^{relabeling}$ **do**
9:             Draw a mini-batch $\mathcal{X}_b$ from $\mathcal{D}_{train}$.
10:           Denoising and relabeling for $\mathcal{X}_b$: solve the Problem (6) by Algorithm 2.
11:         **end for**
12:         Use Eq. 8 to split the training dataset $\mathcal{D}_{train}$ into the clean dataset $\mathcal{D}_{clean}$ and the corrupted dataset $\mathcal{D}_{currupted}$.
13:         **for** $b' = 1, \ldots, N_{batch}^{train}$ **do**
14:            Draw a mini-batch $\mathcal{X}_{b'}$ from $\mathcal{D}_{clean}$, and draw a mini-batch $\mathcal{U}_{b'}$ from $\mathcal{D}_{currupted}$.
15:            **if** $t < T_{sup}$ **then**
16:                $\mathcal{L} = \mathcal{L}_{\mathcal{X}_{b'}}^{mix} + \mathcal{L}_{\mathcal{X}_{b'}}^{lab} + \lambda_1 \mathcal{L}_{\mathcal{U}_{b'}}^{simsiam}$.
17:            **else**
18:                $\mathcal{L} = \mathcal{L}_{\mathcal{X}_{b'}}^{mix} + \mathcal{L}_{\mathcal{X}_{b'}}^{lab} + \lambda_2 \mathcal{L}_{\mathcal{U}_{b'}}^{lab}$.
19:            **end if**
20:            Update model parameter $\theta$ by applying SGD with loss $\mathcal{L}$.
21:         **end for**
22:     **end if**
23: **end for**
24: **Return:** Optimal model parameter $\theta$.

---

We specify our training process in Algorithm S3, which mainly includes two parts, *i.e.* denoising and relabeling part, the training part.

## B Supplement for Experimental Results

### B.1 Comparison with Prediction-, OT- and SOT-Based Pseudo-Labeling

As shown in Fig. S3a and S3b, prediction-based PL generates vague predictions when the class centroids are not discriminative enough. To explain this, prediction-based PL assigns labels in a per-class manner without considering either the global structure of the sample distribution. To this end, OT-based PL optimizes the mapping problem considering the inter-distribution matching of samples and classes, and thus produces more discriminative labels. However, as shown in Fig. S3a, two nearby samples could be mapped to two far-away class centroids, which is not reasonable since it overlooks the inherent coherence structure of the sample distribution, i.e. intra-distribution coherence. Therefore, **our proposed SOT encourages generating more robust labels with both global discriminability and local coherence**.

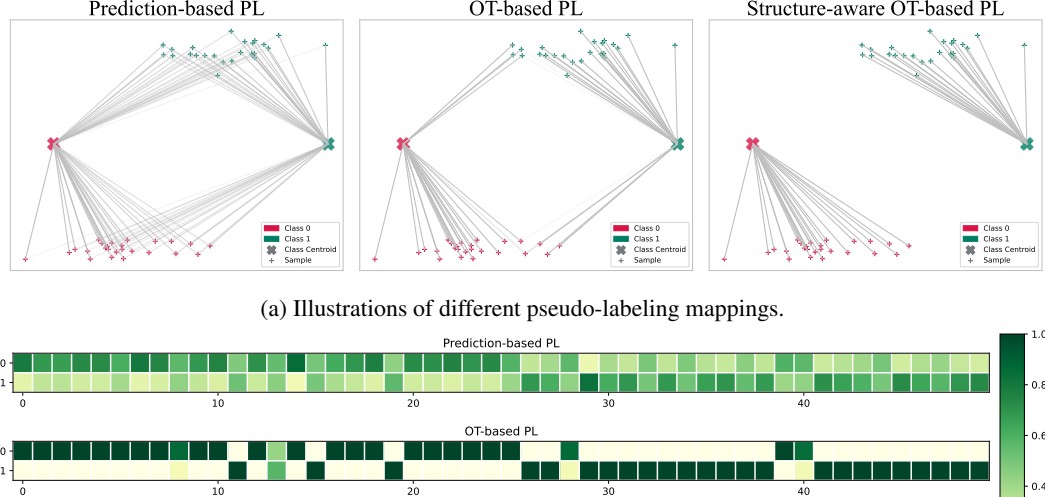

(a) Illustrations of different pseudo-labeling mappings.

(b) Illustrations of different pseudo-labeling (transposed) coupling matrices.

Figure S3: **Comparison with prediction- , OT- and SOT-based pseudo-labeling.** We consider a toy binary classification case for simplicity.

## B.2   Visualization of the Coupling Matrix for CSOT

We visualize randomly selected 200 samples of CIFAR-10 (after 10-epoch warm-up training) and 10 implicit class centroids in feature space in Fig. S4a. The feature dots are visualized based on t-SNE [66], and the implicit class centroids are obtained by a weighted sum of the softmax prediction scores. It is evident that the feature space exhibits confusion in the early training stage, particularly among semantically similar classes, such as cat and dog. Therefore, **utilizing a full mapping based on OT would lead to incorrect assignments for samples that have not yet been sufficiently learned**. Our proposed strategy, on the other hand, demonstrates superiority by selectively assigning reliable labels to a fraction of samples with the highest confidence. This approach ensures high training label accuracy and mitigates the negative impact of unreliable labels during the early stages of training. In addition, we also visualize the coupling matrices, along with their corresponding row and column sum vectors by histograms in Fig. S4b, which illustrates the partial mapping controlled by curriculum constraints.

## B.3   Convergence of the proposed GCG algorithm for CSOT

We set the number of outer loops is set to 10, and the number for inner scaling iteration is set to 100. And the curriculum budget $m$ is set to $0.5$, and the local coherent regularized terms weight $\kappa$ is set to 1. As demonstrated in Fig. S5, our computational method, which includes a novel scaling iteration within a generalized conditional gradient framework, **is capable of optimizing the non-convex objective and converging to a stationary point**.

## B.4   Addictional Results of CSOT

| Method | Time cost |
|---|---|
| DivideMix[46] | 5.1h |
| NCE[45] | 6.5h |
| **CSOT** | **4.8h** |

Table S5: **Comparison of total training time (hours) on CIFAR-10.** The experiments are implemented on a single GPU NVIDIA A100.

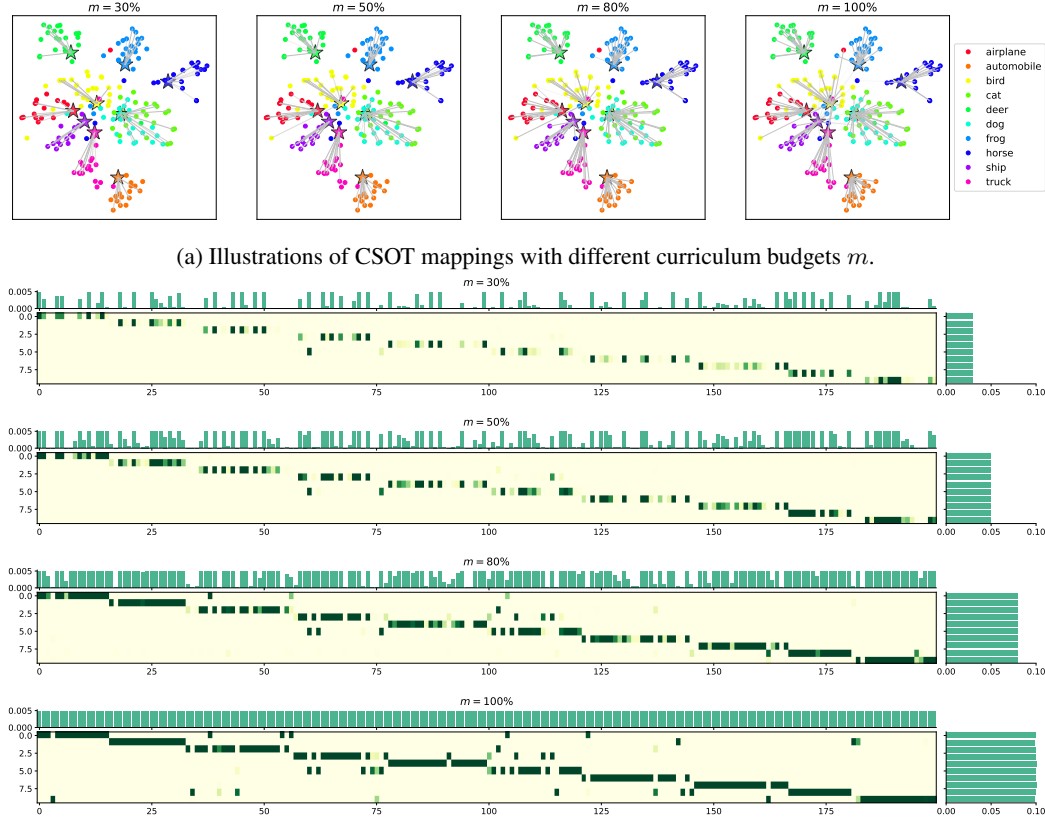

(a) Illustrations of CSOT mappings with different curriculum budgets $m$.

(b) Illustrations of CSOT (transposed) coupling matrices with different curriculum budgets $m$.

Figure S4: **Comparison with using different curriculum budgets $m$.** The samples are plotted as colorful dots and the class centroids are plotted as five-pointed stars, which are colored by their true labels.

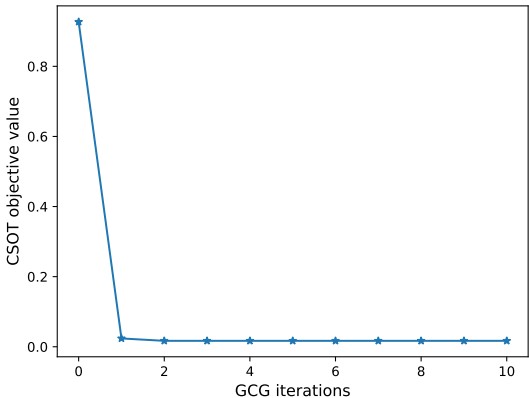

Figure S5: **Convergence behaviour of the generalized conditional gradient (GCG) algorithm for CSOT.**

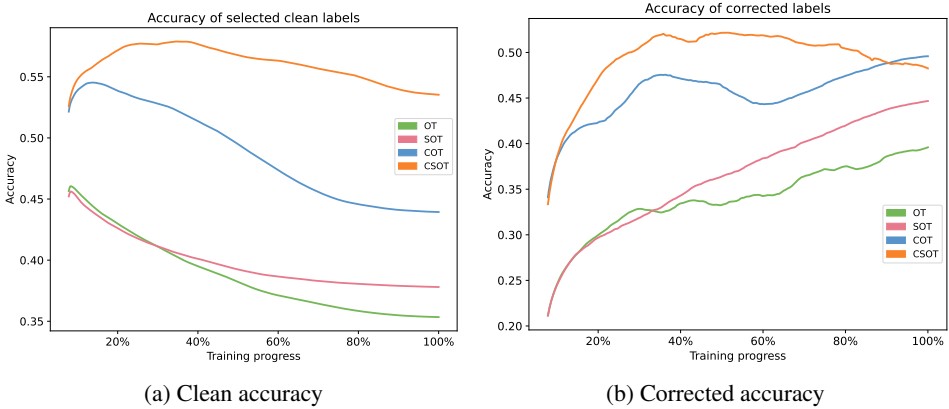

(a) Clean accuracy         (b) Corrected accuracy

Figure S6: **Performance comparison for clean label identification and corrupted label correction.** The experiments are conducted on CIFAR-100 sym0.9.

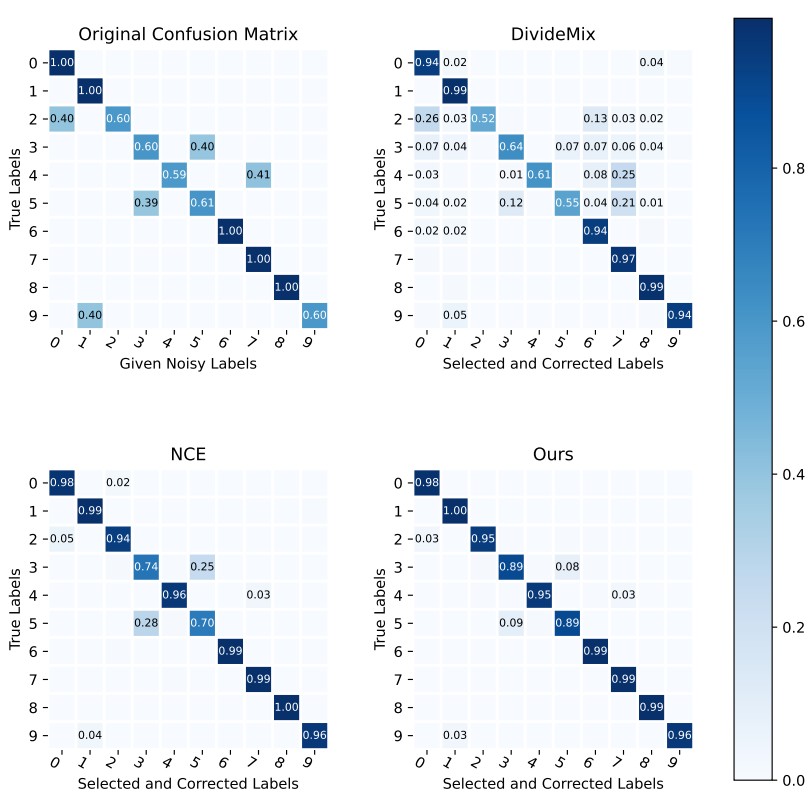

Figure S7: **Comparision of confusion matrix on CIFAR-10 assym-40%.** The darker the color on the diagonal elements of the matrix, the higher the accuracy.

**Effectiveness of CSOT-based denoising and relabeling.** To further verify the effectiveness of our CSOT for clean label identification and corrupted label correction, we also conduct ablation experiments on OT-, SOT-, COT-, and CSOT-based denoising and relabeling. As depicted in Fig. S6, **the incorporation of curriculum constraints ensures high accuracy of clean labels during the early training stage**. This, in turn, facilitates effective learning by providing the model with correct and reliable information, which avoids error accumulation. Furthermore, **local coherent regularized terms contribute to improved label correction**.

Table S6: **Comparison with state-of-the-art methods in test accuracy (%) on Clothing1M.**

| Method | Meta-L. [47] | DivideMix [46] | ELR+ [50] | ELR+ [50] | RRL [48] | NCE‡ [45] | **CSOT** |
|---|---|---|---|---|---|---|---|
| Accuracy | 73.50 | 74.48 | 72.87 | 74.80 | 74.84 | 74.71 | **75.16** |

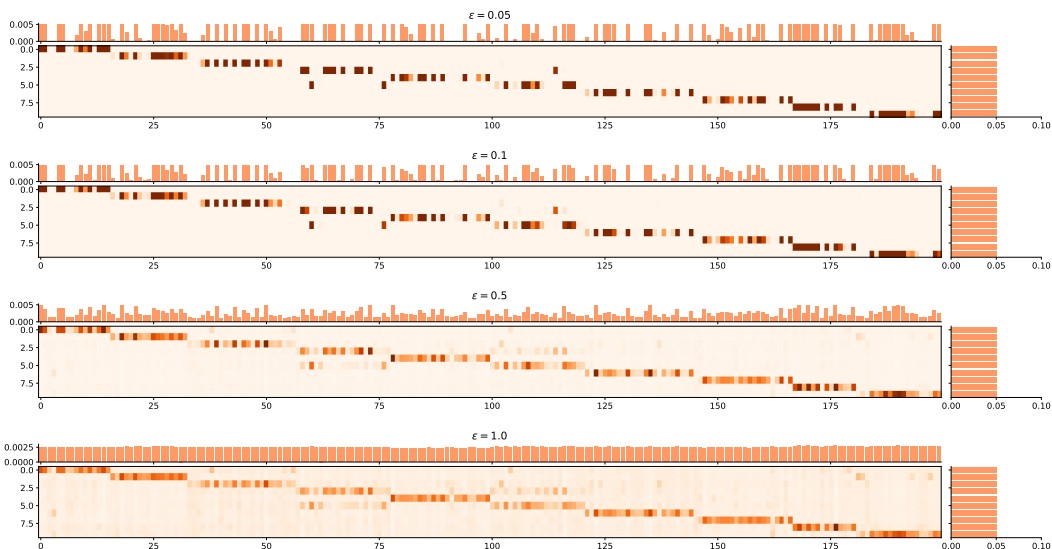

Figure S8: **Visualization of coupling matrix with different entropic regularized weights $\varepsilon$.** We conduct experiments on randomly selected 200 samples of CIFAR-10 (after 10-epoch warm-up training) and the curriculum budget $m$ is set to $0.5$.

**Result of Clothing1M.** Clothing1M [74] is another real-world noisy dataset, which consists of 1 million training images collected from online shopping websites with labels generated from surrounding texts. We use the augmentation provided in [62] as **Aug**($\cdot$). Following the similar setting in NCE [45] and DivideMix [46], we also **conduct the experiment on Clothing1M and achieve superior performance** compared to existing approaches, as shown in Tab. S6. Since NCE utilized an inaccessible data augmentation and hence we reproduce NCE with the augmentation in [62] for a fair comparison, denoted by ‡.

### B.5 Hyperparameter Analysis

**Entropic regularized weight $\varepsilon$.** When $\varepsilon \rightarrow 0$, the entropic regularized CSOT formulation becomes closer to the exact CSOT. Therefore, in order to obtain a solution that closely approximates the exact CSOT, we prefer to set $\varepsilon$ to a small value. We visualize the coupling matrix with different $\varepsilon$ in Fig. S8, and it can be observed that the $\varepsilon$ also influences the mapping smoothness. **A smaller $\varepsilon$ leads to sharper pseudo labels**. To ensure discriminative relabeling and reliable selection, we set $\varepsilon$ to $0.1$.

**Local coherent regularized weight $\kappa$.** The local coherent regularized weight, $\kappa$, determines the strength of local coherent mapping. As shown in Fig. S9, we can observe that the performance is not sensitive to the different values of $\kappa$, and it is relatively easy to tune. It is important to note that **setting $\kappa$ too high can result in performance degradation, particularly in scenarios with high noise rates**. This is because the label-level local consistency term $\Omega^{\mathbf{L}}$, may introduce incorrect consistency in such cases.

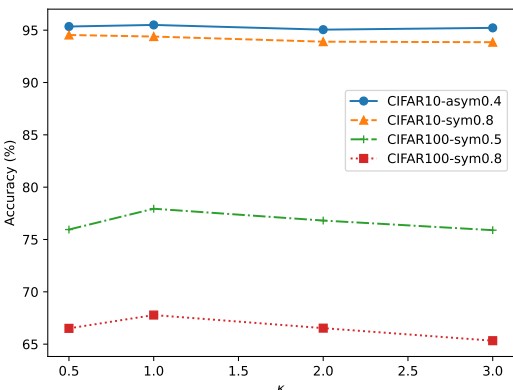

Figure S9: **Sensitivity to the local coherent regularized weight $\kappa$ on four different noisy learning tasks.**

## C  More Discussion about CSOT

## D  Background on Computational Optimal Transport

### D.1  Sinkhorn's Algorithm

---
**Algorithm S4** Sinhorn algorithm, for entropic regularized classical OT
---
1: **Input:** Cost matrix $\mathbf{C}$, marginal constraints vectors $\boldsymbol{\alpha}$ and $\boldsymbol{\beta}$, entropic regularization weight $\varepsilon$
2: Initialize: $\mathbf{K} \leftarrow e^{-\mathbf{C}/\varepsilon}$, $\boldsymbol{v}^{(0)} \leftarrow \mathbb{1}_{|\boldsymbol{\beta}|}$
3: Compute: $\mathbf{K}_{\boldsymbol{\alpha}} \leftarrow \frac{\mathbf{K}}{\text{diag}(\boldsymbol{\alpha})\mathbb{1}_{|\boldsymbol{\alpha}|\times|\boldsymbol{\beta}|}}$, $\mathbf{K}_{\boldsymbol{\beta}}^{\top} \leftarrow \frac{\mathbf{K}^{\top}}{\text{diag}(\boldsymbol{\beta})\mathbb{1}_{|\boldsymbol{\beta}|\times|\boldsymbol{\alpha}|}}$ // Saving computation
4: **for** $n = 1, 2, 3, \ldots$ **do**
5: $\quad \boldsymbol{u}^{(n)} \leftarrow \frac{\mathbb{1}_{|\boldsymbol{\alpha}|}}{\mathbf{K}_{\boldsymbol{\alpha}}\boldsymbol{v}^{(n-1)}}$
6: $\quad \boldsymbol{v}^{(n)} \leftarrow \frac{\mathbb{1}_{|\boldsymbol{\beta}|}}{\mathbf{K}_{\boldsymbol{\beta}}^{\top}\boldsymbol{u}^{(n)}}$
7: **end for**
8: **Return:** $\text{diag}(\boldsymbol{u}^{(n)})\mathbf{K}\text{diag}(\boldsymbol{v}^{(n)})$

---

The Sinkhorn algorithm for solving entropic regularized OT problem is presented in Algorithm S4. It is evident that our proposed scaling iteration is very similar to the existing efficient Sinkhorn algorithm. The main difference lies in Line 5 of Algorithm S4, which corresponds to Line 5 of Algorithm 1. Therefore, our scaling iteration shares the same quadratic time complex as the Sinkhorn algorithm.

### D.2  Relation Between Kullback-Leibler Divergence and Entropic Regularized OT

Given a convex set $\mathcal{C}$, and a matrix $\mathbf{M}$, the projection according to the Kullback-Leibler (KL) divergence is defined as

$$P_{\mathcal{C}}^{\text{KL}}(\mathbf{M}) \overset{\text{def}}{=} \underset{\mathbf{Q}\in\mathcal{C}}{\text{argmin}}\, \text{KL}(\mathbf{Q}|\mathbf{M}). \tag{S22}$$

According to [8], the classical OT can be rewritten in a KL projection form as follows:

$$\min_{\mathbf{Q}\in\mathbf{\Pi}(\boldsymbol{\alpha},\boldsymbol{\beta})} \langle \mathbf{C}, \mathbf{Q}\rangle = \min_{\mathbf{Q}\in\mathbf{\Pi}(\boldsymbol{\alpha},\boldsymbol{\beta})} \varepsilon\text{KL}(\mathbf{Q}|e^{-\mathbf{C}/\varepsilon}), \tag{S23}$$

which can be interpreted as that solving a classical OT problem is equivalent to solving a KL projection from a given matrix $e^{-\mathbf{C}/\varepsilon}$ to the constraint $\mathbf{\Pi}(\boldsymbol{\alpha},\boldsymbol{\beta})$. In light of this, it was proposed in [8] that when $\mathcal{C}$ is an intersection of closed convex and affine sets, the classical OT problem can be solved by iterative Bregman projections [10]. However, when $\mathcal{C}$ is an intersection of closed convex

but not affine sets, Dykstra's algorithm [21] is employed to guarantee convergence [7], as iterative Bregman projections do not generally converge to the KL projection on the intersection.

## D.3 Dykstra's Algorithm

Assume that $\mathcal{C}$ is an intersection of closed convex but not affine sets:

$$\mathcal{C} = \bigcap_{\ell=1}^{L} \mathcal{C}_\ell, \tag{S24}$$

and we extend the indexing of the sets by $L$-periodicity so that they satisfy

$$\forall n \in \mathbb{N}, \quad \mathcal{C}_{n+L} = \mathcal{C}_n. \tag{S25}$$

Dykstra's algorithm [21] starts by initializing

$$\mathbf{Q}^{(0)} = \mathbf{K} \quad \text{and} \quad \mathbf{U}^{(0)} = \mathbf{U}^{(-1)} = \cdots = \mathbf{U}^{(-L+1)} = \mathbb{1}. \tag{S26}$$

One then iteratively defines

$$\mathbf{Q}^{(n)} = P_{\mathcal{C}_n}^{\mathrm{KL}}(\mathbf{Q}^{(n-1)} \odot \mathbf{U}^{(n-L)}), \quad \text{and} \quad \mathbf{U}^{(n)} = \mathbf{U}^{(n-L)} \odot \frac{\mathbf{Q}^{(n-1)}}{\mathbf{Q}^{(n)}}. \tag{S27}$$

## D.4 Generalized Conditional Gradient Algorithm

We are interested in the problem of minimizing under constraints a composite function such as

$$\min_{\mathbf{Q} \in \mathcal{C}} = f(\mathbf{Q}) + g(\mathbf{Q}), \tag{S28}$$

where both $f(\cdot)$ is a differentiable and possibly non-convex function; $g(\cdot)$ is a convex, possibly non-differentiable function; $\mathcal{C}$ denotes any convex and compact set. One might want to benefit from this composite structure during the optimization procedure. For instance, if we have an efficient solver for optimizing

$$\min_{\mathbf{Q} \in \mathcal{C}} = \langle \nabla f, \mathbf{Q} \rangle + g(\mathbf{Q}). \tag{S29}$$

It is of prime interest to use this solver in the optimization scheme instead of linearizing the whole objective function as one would do with a conditional gradient algorithm [9, 59], as shown in Algorithm S5.

---

**Algorithm S5** Generalized conditional gradient algorithm

---

1: **Input:** A differentiable and possibly non-convex function $f$ and its gradient function $\nabla f$, a convex, possibly non-differentiable function $g$, a convex and compact set $\mathcal{C}$.
2: Initialize: $\mathbf{Q}^{(0)} \in \mathcal{C}$
3: **for** $i = 1, 2, 3, \ldots$ **do**
4:   $\mathbf{G}^{(i)} \leftarrow \mathbf{Q}^{(i)} + \nabla f(\mathbf{Q}^{(i)})$ // Gradient computation
5:   $\widetilde{\mathbf{Q}}^{(i)} \leftarrow \mathrm{argmin}_{\mathbf{Q} \in \mathcal{C}} \langle \mathbf{Q}, \mathbf{G}^{(i)} \rangle + g(\mathbf{Q})$ // Partial linearization
6:   Find the optimal step $\eta^{(i)}$ with $\Delta \mathbf{Q} = \widetilde{\mathbf{Q}}^{(i)} - \mathbf{Q}^{(i)}$

$$\eta^{(i)} = \underset{\eta \in [0,1]}{\mathrm{argmin}} \, f(\mathbf{Q}^{(i)} + \eta^{(i)} \Delta \mathbf{Q}) + g(\mathbf{Q}^{(i)} + \eta^{(i)} \Delta \mathbf{Q})$$

   or choose $\eta^{(i)} \in [0, 1]$ so that it satisfies the Armijo rule.
   // Exact or backtracking line-search
7:   $\mathbf{Q}^{(i+1)} \leftarrow (1 - \eta^{(i)}) \mathbf{Q}^{(i)} + \eta^{(i)} \widetilde{\mathbf{Q}}^{(i)}$ // Update
8: **end for**
9: **Return:** $\mathbf{Q}^{(i)}$

---

# E  Derivation Details of the Efficient Scaling Iteration Method (Lemma 1)

We have developed a lightspeed computational method that involves a scaling iteration within a generalized conditional gradient framework to solve CSOT efficiently. Specifically, the efficiency is mainly brought by the scaling iteration method for solving the COT problem (Problem (12)), which is proposed in Lemma 1.

This section presents the derivation details of this efficient scaling iteration method. First, we show that solving COT is equivalent to solving the KL projection problem with the curriculum constraints (Lemma S2). Then such a KL projection problem can be solved by iterating Dykstras algorithm (Lemma S3). However, Dykstras algorithm is based on matrix-matrix multiplication which is computationally extensive. Therefore, we propose a fast implementation of Dykstras algorithm by only performing matrix-vector multiplications, *i.e.* efficient scaling iteration (Lemma 1).

**Lemma S2.** *Solving the Problem (12) is equivalent to solving the KL projection problem from the matrix $e^{-\mathbf{C}/\varepsilon}$ to the curriculum constraint $\mathbf{\Pi}^c(\boldsymbol{\alpha}, \boldsymbol{\beta})$, i.e.*

$$\min_{\mathbf{Q} \in \mathbf{\Pi}^c(\boldsymbol{\alpha},\boldsymbol{\beta})} \langle \mathbf{C}, \mathbf{Q} \rangle + \varepsilon \langle \mathbf{Q}, \log \mathbf{Q} \rangle \Leftrightarrow \min_{\mathbf{Q} \in \mathbf{\Pi}^c(\boldsymbol{\alpha},\boldsymbol{\beta})} \varepsilon KL(\mathbf{Q}|e^{-\mathbf{C}/\varepsilon}), \tag{S30}$$

*Proof.*

$$\min_{\mathbf{Q} \in \mathbf{\Pi}^c(\boldsymbol{\alpha},\boldsymbol{\beta})} \langle \mathbf{C}, \mathbf{Q} \rangle + \varepsilon \langle \mathbf{Q}, \log \mathbf{Q} \rangle$$

$$= \min_{\mathbf{Q} \in \mathbf{\Pi}^c(\boldsymbol{\alpha},\boldsymbol{\beta})} \langle \mathbf{Q}, \mathbf{C} + \varepsilon \log \mathbf{Q} \rangle$$

$$= \min_{\mathbf{Q} \in \mathbf{\Pi}^c(\boldsymbol{\alpha},\boldsymbol{\beta})} \varepsilon \langle \mathbf{Q}, \mathbf{C}/\varepsilon + \log \mathbf{Q} \rangle$$

$$= \min_{\mathbf{Q} \in \mathbf{\Pi}^c(\boldsymbol{\alpha},\boldsymbol{\beta})} \varepsilon \left\langle \mathbf{Q}, \log \frac{\mathbf{Q}}{e^{-\mathbf{C}/\varepsilon}} \right\rangle$$

$$= \min_{\mathbf{Q} \in \mathbf{\Pi}^c(\boldsymbol{\alpha},\boldsymbol{\beta})} \varepsilon KL(\mathbf{Q}|e^{-\mathbf{C}/\varepsilon})$$

□

Recall that the curriculum constraints $\mathbf{\Pi}^c(\boldsymbol{\alpha}, \boldsymbol{\beta})$ can be expressed as an intersection of two convex but not affine sets:

$$\mathcal{C}_1 \stackrel{\text{def}}{=} \left\{ \mathbf{Q} \in \mathbb{R}_+^{|\boldsymbol{\alpha}| \times |\boldsymbol{\beta}|} | \mathbf{Q}\mathbb{1}_{|\boldsymbol{\beta}|} \leq \boldsymbol{\alpha} \right\} \quad \text{and} \quad \mathcal{C}_2 \stackrel{\text{def}}{=} \left\{ \mathbf{Q} \in \mathbb{R}_+^{|\boldsymbol{\alpha}| \times |\boldsymbol{\beta}|} | \mathbf{Q}^\top \mathbb{1}_{|\boldsymbol{\alpha}|} = \boldsymbol{\beta} \right\}. \tag{S31}$$

**Lemma S3.** *The KL projection from a matrix $\mathbf{M}$ to the $\mathcal{C}_1$ and $\mathcal{C}_2$ are expressed as*

$$P_{\mathcal{C}_1}^{KL}(\mathbf{M}) = \text{diag}\left(\min\left(\frac{\boldsymbol{\alpha}}{\mathbf{M}\mathbb{1}_{|\boldsymbol{\beta}|}}, \mathbb{1}_{|\boldsymbol{\beta}|}\right)\right)\mathbf{M}, \tag{S32}$$

$$P_{\mathcal{C}_2}^{KL}(\mathbf{M}) = \mathbf{M}\text{diag}\left(\frac{\boldsymbol{\beta}}{\mathbf{M}^\top \mathbb{1}_{|\boldsymbol{\alpha}|}}\right). \tag{S33}$$

*Then the Problem (12) can be solved by Dykstra iterations, presented in Algorithm S6.*

Lemma S3 can be derived form the Proposition 1 and Proposition 5 in [8].

The limitation of Dykstras algorithm comes from its computationally extensive matrix-matrix multiplication. To handle this issue, we propose a fast implementation of Dykstras algorithm by only performing matrix-vector multiplications, *i.e.* efficient scaling iteration (Lemma 1).

**Lemma 1.** *(Efficient scaling iteration for Curriculum OT) When solving Problem (12) by iterating Dykstra's algorithm, the matrix $\mathbf{Q}^{(n)}$ at $n$ iteration is a diagonal scaling of $\mathbf{K} := e^{-\mathbf{C}/\varepsilon}$, which is the element-wise exponential matrix of $-\mathbf{C}/\varepsilon$:*

$$\mathbf{Q}^{(n)} = \text{diag}\left(\boldsymbol{u}^{(n)}\right)\mathbf{K}\text{diag}\left(\boldsymbol{v}^{(n)}\right), \tag{S34}$$

*where the vectors $\boldsymbol{u}^{(n)} \in \mathbb{R}^{|\boldsymbol{\alpha}|}$, $\boldsymbol{v}^{(n)} \in \mathbb{R}^{|\boldsymbol{\beta}|}$ satisfy $\boldsymbol{v}^{(0)} = \mathbb{1}_{|\boldsymbol{\beta}|}$ and follow the recursion formula*

$$\boldsymbol{u}^{(n)} = \min\left(\frac{\boldsymbol{\alpha}}{\mathbf{K}\boldsymbol{v}^{(n-1)}}, \mathbb{1}_{|\boldsymbol{\alpha}|}\right) \quad \text{and} \quad \boldsymbol{v}^{(n)} = \frac{\boldsymbol{\beta}}{\mathbf{K}^\top \boldsymbol{u}^{(n)}}. \tag{S35}$$

**Algorithm S6** Dykstras algorithm for entropic regularized Curriculum OT

---

1: **Input:** Cost matrix $\mathbf{C}$, marginal constraints vectors $\boldsymbol{\alpha}$ and $\boldsymbol{\beta}$, entropic regularization weight $\varepsilon$
2: Initialize: $\mathbf{Q}^{(0)} \leftarrow e^{-\mathbf{C}/\varepsilon}$, $\mathbf{U}'^{(0)} \leftarrow \mathbb{1}_{|\boldsymbol{\alpha}| \times |\boldsymbol{\beta}|}$, $\mathbf{U}^{(0)} \leftarrow \mathbb{1}_{|\boldsymbol{\alpha}| \times |\boldsymbol{\beta}|}$
3: **for** $t = 1, 2, 3, \ldots$ **do**
4:     $\mathbf{Q}'^{(t)} \leftarrow P_{\mathcal{C}_1}^{\mathrm{KL}}(\mathbf{Q}^{(t-1)} \odot \mathbf{U}'^{(t-1)})$
5:     $\mathbf{U}'^{(t)} \leftarrow \mathbf{U}'^{(t-1)} \odot \frac{\mathbf{Q}^{(t-1)}}{\mathbf{Q}'^{(t)}}$
6:     $\mathbf{Q}^{(t)} \leftarrow P_{\mathcal{C}_2}^{\mathrm{KL}}(\mathbf{Q}'^{(t)} \odot \mathbf{U}^{(t-1)})$
7:     $\mathbf{U}^{(t)} \leftarrow \mathbf{U}^{(t-1)} \odot \frac{\mathbf{Q}'^{(t)}}{\mathbf{Q}^{(t)}}$
8: **end for**
9: **Return:** $\mathbf{Q}^{(t)}$

---

*Proof.* Firstly, let $\boldsymbol{u}^{(1)} := \min\left(\frac{\boldsymbol{\alpha}}{\mathbf{Q}^{(0)}\mathbb{1}_{|\boldsymbol{\beta}|}}, \mathbb{1}_{|\boldsymbol{\alpha}|}\right)$. Following the Algorithm S6 and Lemma S3, we derive $\mathbf{Q}'^{(1)}$ and $\mathbf{U}'^{(1)}$. Now we have

$$\mathbf{Q}'^{(1)} = P_{\mathcal{C}_1}^{\mathrm{KL}}(\mathbf{Q}^{(0)} \odot \mathbf{U}'^{(0)}) = \mathrm{diag}\left(\min\left(\frac{\boldsymbol{\alpha}}{\mathbf{Q}^{(0)}\mathbb{1}_{|\boldsymbol{\beta}|}}, \mathbb{1}_{|\boldsymbol{\alpha}|}\right)\right)\mathbf{Q}^{(0)} = \mathrm{diag}\left(\boldsymbol{u}^{(1)}\right)\mathbf{Q}^{(0)},$$

$$\mathbf{U}'^{(1)} = \mathbf{U}'^{(0)} \odot \frac{\mathbf{Q}^{(0)}}{\mathbf{Q}'^{(1)}} = \frac{\mathbf{Q}^{(0)}}{\mathrm{diag}\left(\boldsymbol{u}^{(1)}\right)\mathbf{Q}^{(0)}} = \mathrm{diag}\left(1/\boldsymbol{u}^{(1)}\right)\mathbb{1}_{|\boldsymbol{\alpha}| \times |\boldsymbol{\beta}|}.$$

Then let $\boldsymbol{v}^{(1)} := \frac{\boldsymbol{\beta}}{\mathbf{Q}^{(0)\top}\boldsymbol{u}^{(1)}}$. And we derive $\mathbf{Q}^{(1)}$ and $\mathbf{U}^{(1)}$ as follows:

$$
\begin{aligned}
\mathbf{Q}^{(1)} &= P_{\mathcal{C}_2}^{\mathrm{KL}}(\mathbf{Q}'^{(1)} \odot \mathbf{U}^{(0)}) \\
&= \mathbf{Q}'^{(1)}\mathrm{diag}\left(\frac{\boldsymbol{\beta}}{\mathbf{K}^{(1)\top}\mathbb{1}_{|\boldsymbol{\alpha}|}}\right) \\
&= \mathrm{diag}\left(\boldsymbol{u}^{(1)}\right)\mathbf{Q}^{(0)}\mathrm{diag}\left(\frac{\boldsymbol{\beta}}{\mathbf{Q}^{(0)\top}\mathrm{diag}\left(\boldsymbol{u}^{(1)}\right)\mathbb{1}_{|\boldsymbol{\alpha}|}}\right) \\
&= \mathrm{diag}\left(\boldsymbol{u}^{(1)}\right)\mathbf{Q}^{(0)}\mathrm{diag}\left(\frac{\boldsymbol{\beta}}{\mathbf{Q}^{(0)\top}\boldsymbol{u}^{(1)}}\right) \\
&= \mathrm{diag}\left(\boldsymbol{u}^{(1)}\right)\mathbf{Q}^{(0)}\mathrm{diag}\left(\boldsymbol{v}^{(1)}\right),
\end{aligned}
$$

$$\mathbf{U}^{(1)} = \mathbf{U}^{(0)} \odot \frac{\mathbf{Q}'^{(1)}}{\mathbf{Q}^{(1)}} = \frac{\mathrm{diag}\left(\boldsymbol{u}^{(1)}\right)\mathbf{Q}^{(0)}}{\mathrm{diag}\left(\boldsymbol{u}^{(1)}\right)\mathbf{Q}^{(0)}\mathrm{diag}\left(\boldsymbol{v}^{(1)}\right)} = \mathbb{1}_{|\boldsymbol{\alpha}| \times |\boldsymbol{\beta}|}\mathrm{diag}\left(1/\boldsymbol{v}^{(1)}\right).$$

For simplicity, before deriving $\mathbf{Q}'^{(2)}$ and $\mathbf{U}'^{(2)}$, we derive $\mathbf{Q}^{(1)} \odot \mathbf{U}'^{(1)}$ firstly:

$$
\begin{aligned}
\mathbf{Q}^{(1)} \odot \mathbf{U}'^{(1)} &= \left(\mathrm{diag}\left(\boldsymbol{u}^{(1)}\right)\mathbf{Q}^{(0)}\mathrm{diag}\left(\boldsymbol{v}^{(1)}\right)\right) \odot \left(\mathrm{diag}\left(1/\boldsymbol{u}^{(1)}\right)\mathbb{1}_{|\boldsymbol{\alpha}| \times |\boldsymbol{\beta}|}\right) \\
&= \left(\mathbf{Q}^{(0)}\mathrm{diag}\left(\boldsymbol{v}^{(1)}\right)\right) \odot \left(\mathbb{1}_{|\boldsymbol{\alpha}| \times |\boldsymbol{\beta}|}\right) \\
&= \mathbf{Q}^{(0)}\mathrm{diag}\left(\boldsymbol{v}^{(1)}\right).
\end{aligned}
$$

Let $\boldsymbol{u}^{(2)} := \min\left(\frac{\boldsymbol{\alpha}}{\mathbf{Q}^{(0)}\boldsymbol{v}^{(1)}}, \mathbb{1}_{|\boldsymbol{\alpha}|}\right)$. We can now derive $\mathbf{Q}'^{(2)}$ and $\mathbf{U}'^{(2)}$ as follows:

$$
\begin{aligned}
\mathbf{Q}'^{(2)} &= P^{\mathrm{KL}}_{\mathcal{C}_1}(\mathbf{Q}^{(1)} \odot \mathbf{U}'^{(1)}) \\
&= \operatorname{diag}\left(\min\left(\frac{\boldsymbol{\alpha}}{\left(\mathbf{Q}^{(1)} \odot \mathbf{U}'^{(1)}\right)\mathbb{1}_{|\boldsymbol{\beta}|}}, \mathbb{1}_{|\boldsymbol{\alpha}|}\right)\right)\left(\mathbf{Q}^{(1)} \odot \mathbf{U}'^{(1)}\right) \\
&= \operatorname{diag}\left(\min\left(\frac{\boldsymbol{\alpha}}{\mathbf{Q}^{(0)}\operatorname{diag}\left(\boldsymbol{v}^{(1)}\right)\mathbb{1}_{|\boldsymbol{\beta}|}}, \mathbb{1}_{|\boldsymbol{\alpha}|}\right)\right)\mathbf{Q}^{(0)}\operatorname{diag}\left(\boldsymbol{v}^{(1)}\right) \\
&= \operatorname{diag}\left(\min\left(\frac{\boldsymbol{\alpha}}{\mathbf{Q}^{(0)}\boldsymbol{v}^{(1)}}, \mathbb{1}_{|\boldsymbol{\alpha}|}\right)\right)\mathbf{Q}^{(0)}\operatorname{diag}\left(\boldsymbol{v}^{(1)}\right) \\
&= \operatorname{diag}\left(\boldsymbol{u}^{(2)}\right)\mathbf{Q}^{(0)}\operatorname{diag}\left(\boldsymbol{v}^{(1)}\right),
\end{aligned}
$$

$$
\begin{aligned}
\mathbf{U}'^{(2)} &= \mathbf{U}'^{(1)} \odot \frac{\mathbf{Q}^{(1)}}{\mathbf{Q}'^{(2)}} \\
&= \left(\operatorname{diag}\left(1/\boldsymbol{u}^{(1)}\right)\mathbb{1}_{|\boldsymbol{\alpha}|\times|\boldsymbol{\beta}|}\right) \odot \frac{\operatorname{diag}\left(\boldsymbol{u}^{(1)}\right)\mathbf{Q}^{(0)}\operatorname{diag}\left(\boldsymbol{v}^{(1)}\right)}{\operatorname{diag}\left(\boldsymbol{u}^{(2)}\right)\mathbf{Q}^{(0)}\operatorname{diag}\left(\boldsymbol{v}^{(1)}\right)} \\
&= \left(\operatorname{diag}\left(1/\boldsymbol{u}^{(1)}\right)\mathbb{1}_{|\boldsymbol{\alpha}|\times|\boldsymbol{\beta}|}\right) \odot \frac{\operatorname{diag}\left(\boldsymbol{u}^{(1)}\right)}{\operatorname{diag}\left(\boldsymbol{u}^{(2)}\right)} \\
&= \left(\operatorname{diag}\left(1/\boldsymbol{u}^{(1)}\right)\mathbb{1}_{|\boldsymbol{\alpha}|\times|\boldsymbol{\beta}|}\right) \odot \operatorname{diag}\left(\boldsymbol{u}^{(1)}/\boldsymbol{u}^{(2)}\right) \\
&= \operatorname{diag}\left(1/\boldsymbol{u}^{(2)}\right)\mathbb{1}_{|\boldsymbol{\alpha}|\times|\boldsymbol{\beta}|}.
\end{aligned}
$$

For simplicity, before deriving $\mathbf{Q}^{(2)}$ and $\mathbf{U}^{(2)}$, we derive $\mathbf{Q}'^{(2)} \odot \mathbf{U}^{(1)}$ firstly:

$$
\begin{aligned}
\mathbf{Q}'^{(2)} \odot \mathbf{U}^{(1)} &= \left(\operatorname{diag}\left(\boldsymbol{u}^{(2)}\right)\mathbf{Q}^{(0)}\operatorname{diag}\left(\boldsymbol{v}^{(1)}\right)\right) \odot \left(\mathbb{1}_{|\boldsymbol{\alpha}|\times|\boldsymbol{\beta}|}\operatorname{diag}\left(1/\boldsymbol{v}^{(1)}\right)\right) \\
&= \left(\operatorname{diag}\left(\boldsymbol{u}^{(2)}\right)\mathbf{Q}^{(0)}\right) \odot \mathbb{1}_{|\boldsymbol{\alpha}|\times|\boldsymbol{\beta}|} \\
&= \operatorname{diag}\left(\boldsymbol{u}^{(2)}\right)\mathbf{Q}^{(0)}.
\end{aligned}
$$

Let $\boldsymbol{v}^{(2)} := \frac{\boldsymbol{\beta}}{\mathbf{Q}^{(0)\top}\boldsymbol{u}^{(2)}}$. We can now derive $\mathbf{Q}^{(2)}$ and $\mathbf{U}^{(2)}$ as follows:

$$
\begin{aligned}
\mathbf{Q}^{(2)} &= P^{\mathrm{KL}}_{\mathcal{C}_2}(\mathbf{Q}'^{(2)} \odot \mathbf{U}^{(1)}) \\
&= \operatorname{diag}\left(\boldsymbol{u}^{(2)}\right)\mathbf{Q}^{(0)}\operatorname{diag}\left(\frac{\boldsymbol{\beta}}{\left(\operatorname{diag}\left(\boldsymbol{u}^{(2)}\right)\mathbf{Q}^{(0)}\right)^{\top}\mathbb{1}_{|\boldsymbol{\alpha}|}}\right) \\
&= \operatorname{diag}\left(\boldsymbol{u}^{(2)}\right)\mathbf{Q}^{(0)}\operatorname{diag}\left(\frac{\boldsymbol{\beta}}{\mathbf{Q}^{(0)\top}\boldsymbol{u}^{(2)}}\right) \\
&= \operatorname{diag}\left(\boldsymbol{u}^{(2)}\right)\mathbf{Q}^{(0)}\operatorname{diag}\left(\boldsymbol{v}^{(2)}\right)
\end{aligned}
$$

$$
\begin{aligned}
\mathbf{U}^{(2)} &= \mathbf{U}^{(1)} \odot \frac{\mathbf{Q}'^{(2)}}{\mathbf{Q}^{(2)}} \\
&= \left(\mathbb{1}_{|\boldsymbol{\alpha}|\times|\boldsymbol{\beta}|}\operatorname{diag}\left(1/\boldsymbol{v}^{(1)}\right)\right) \odot \frac{\operatorname{diag}\left(\boldsymbol{u}^{(2)}\right)\mathbf{Q}^{(0)}\operatorname{diag}\left(\boldsymbol{v}^{(1)}\right)}{\operatorname{diag}\left(\boldsymbol{u}^{(2)}\right)\mathbf{Q}^{(0)}\operatorname{diag}\left(\boldsymbol{v}^{(2)}\right)} \\
&= \left(\mathbb{1}_{|\boldsymbol{\alpha}|\times|\boldsymbol{\beta}|}\operatorname{diag}\left(1/\boldsymbol{v}^{(1)}\right)\right) \odot \frac{\operatorname{diag}\left(\boldsymbol{v}^{(1)}\right)}{\operatorname{diag}\left(\boldsymbol{v}^{(2)}\right)} \\
&= \left(\mathbb{1}_{|\boldsymbol{\alpha}|\times|\boldsymbol{\beta}|}\operatorname{diag}\left(1/\boldsymbol{v}^{(1)}\right)\right) \odot \operatorname{diag}\left(\boldsymbol{v}^{(1)}/\boldsymbol{v}^{(2)}\right) \\
&= \mathbb{1}_{|\boldsymbol{\alpha}|\times|\boldsymbol{\beta}|}\operatorname{diag}\left(1/\boldsymbol{v}^{(2)}\right)
\end{aligned}
$$

To conclude, it can be easily summarized that

$$\mathbf{Q}^{(n)} = \operatorname{diag}\left(\boldsymbol{u}^{(n)}\right)\mathbf{K}\operatorname{diag}\left(\boldsymbol{v}^{(n)}\right),$$

where $\boldsymbol{u}^{(n)} = \min\left(\frac{\boldsymbol{\alpha}}{\mathbf{K}\boldsymbol{v}^{(n-1)}}, \mathbb{1}_{|\boldsymbol{\alpha}|}\right)$, $\boldsymbol{v}^{(n)} = \frac{\boldsymbol{\beta}}{\mathbf{K}^{\top}\boldsymbol{u}^{(n)}}$, and $\boldsymbol{v}^{(0)} = \mathbb{1}_{|\boldsymbol{\beta}|}$.

$\square$