# OpenReview forum: "CSOT: Curriculum and Structure-Aware Optimal Transport for Learning with Noisy Labels"
_NeurIPS.cc/2023/Conference — NeurIPS 2023 poster_

### Official Review · Reviewer_7s92 · 2023-07-02

**Soundness:** 3 good
**Presentation:** 4 excellent
**Contribution:** 3 good
**Rating:** 6
**Confidence:** 5

**Summary:**

The paper studies the problem of noisy label learning. The paper adopts an optimal transport approach to generate pseudo labels for noisy samples. Particularly, the paper builds on the existing method and adds additional regularization terms to enforce the consistency between sample classes and learned representations. The paper also extends the sinkhorn algorithm to solve the proposed OT objective efficiently. Empirically, the proposed method has improved performance over baselines on widely used datasets with various noisy ratios.

**Strengths:**

1. The paper extends the existing optimal transport approach to include the consistency between the sample representations and predictions/labels. It is a novel objective and a solid idea intuitively.

2. The paper extends the sinkhorn algorithm to solve the proposed new OT objective efficiently.

3. The proposed method has strong empirical performance, especially for high noise ratios.

4. As a pseudo-labeling step, the proposed method can potentially work with other noisy label learning objectives.

**Weaknesses:**

1. The proposed method is a regularization of the existing optimal transport pseudo-labeling method. The novelty is thus limited.

2. The introduced regularization uses the same weight kappa for the two terms, while the two terms could have quite different behaviors/values.

3. It is not clear to me how does the method perform or should be modified in the case where the class distribution is imbalanced.

4. The OT objective is not directly related to the training objective but serves as a sieving step. It would be great if a more comprehensive training objective can be formalized to include the OT based selection.

**Questions:**

1. Please address the weaknesses 2 3 and 4.

2. What's an explanation for the experimental results that the given method does better in top-1 ACC but worse in top-5 for the imagenet and webvision datasets?

3. The global relationship modeled by OT depends on the batch size. How does the batch size affect the empirical performance? How does the number of total classes in the training set affect the choice of the ideal batch size for the OT step?

**Limitations:**

Yes.

---

> ### Author Rebuttal · Authors · 2023-08-09
>
> We thank the reviewer for the positive feedback and constructive suggestions on our paper. We address your detailed comments below:
>
> > **Q1**. the proposed method is a regularization of the existing OT pseudo-labeling (PL) method. The novelty is thus limited.
>
> **A1**. Firstly, we would like to justify that our CSOT formulation not only involves two local coherent regularization terms but also curriculum constraints. Notably, **CSOT introduces a novel OT formulation, necessitating a fresh solver. Thus, we devise an lightspeed computational method**.
>
> Secondly, we emphasize that directly applying the original OT to certain PL tasks may result in sub-optimal performance. Consequently, it becomes necessary for researchers to investigate more adaptive OT formulations for specific problems [9][54], including off-the-shelf variants like unbalanced OT and partial OT, to address the specific challenges of the problem. In this context, we propose a novel CSOT formulation tailored for the denoising and relabeling task.
>
> Thirdly, we highlight that our work serves as a valuable example of developing a customized OT formulation and a corresponding solver. By showcasing our method's adaptability and efficacy, we aim to contribute to the wider application and exploration of OT across various domains.
>
>
> > **Q2**. The introduced regularization uses the same weight kappa for the two terms, while the two terms could have quite different behaviors/values.
>
> **A2**. We agree with the reviewer that there may exist better $\kappa_1$ and $\kappa_2$ for $\Omega^P$ and $\Omega^L$ accordingly in Eq.(3). However, practically, it is rather cumbersome to tune separate hyperparameters for different datasets or noise rates, and we set a unified $\kappa$ for simplicity.
>
> Moreover, we find the $\kappa_1$ and $\kappa_2$ are not very sensitive as shown in the Table G4 (global response pdf). The table also reveals that in scenarios with high noise rates like CIFAR-100 sym-0.8, prioritizing the prediction-level term with a higher weight is more advantageous due to label unreliability.
>
>
> > **Q3**. How does the method perform or should be modified in the case where the class distribution is imbalanced?
>
> **A3**. Thank you for raising the concern. Following existing work [18][54], like SwAV, we adopt a uniform class marginal vector for simplicity. While we do not address the class imbalance in this paper within the scope of LNL, we believe our approach can be extended for this purpose in future work. Here's a possible solution: we can introduce an outer minimizer to optimize an appropriate imbalanced class distribution $\beta$:
> $\min_\beta \min_{Q\in\Pi^c(\alpha,\beta)}
>             \left<C, Q\right>
>         +\kappa\Omega(Q)
>         +\varepsilon\left<Q, \log Q\right>$
>
> s.t. $\beta \in $ {$ \beta\in\mathbb{R}_{+}^{C} | \sum_i^C \beta_i=m $}.
>
>
> > **Q4**. The OT objective is not directly related to the training objective but serves as a sieving step. It would be great if a more comprehensive training objective can be formalized to include the OT-based selection.
>
> **A4**. Thank you for your constructive suggestion.
>
> Firstly, we claim that an additional OT-based training objective isn't essential. To explain this, our CSOT-based Pseudo-Labeling (PL) aims at selecting top-confident samples and relabeling them with reliable labels. Following existing work [18][54], reliable labels from OT-based PL seamlessly plug into diverse supervised or semi-supervised losses. Thus, CSOT-based denoising and relabeling, followed by training with standard objectives, offers flexibility.
>
> Secondly, our offline sieving optimization, unsuitable for model training. To be more specific, following DivideMix and NCE, our learning scheme is based on semi-supervised learning as shown in Algorithm S3 (Appendix A.3). And the sieving step based on CSOT has to be offline to get the labeled and unlabeled dataset for the follow-up semi-supervised learning step. Therefore, it is able to learn from the selected and relabeled samples based on off-the-shelf training objectives instead of the offline sieving objective.
>
>
> > **Q5**. The explanation for the experimental results that the given method does better in top-1 ACC but worse in top-5 for the Imagenet and WebVision datasets.
>
> **A5**. Thank you for raising the concern. There are two potential reasons behind this.
>
> Firstly, we reckon that the class-imbalance scenario in the Webvision dataset can not be fully well-addressed in our work. As mentioned in Section 7, class-imbalance cases are specifically not considered in our work and hence we simply adopt uniform distribution as the categories distribution. The uniform distribution helps CSOT allocate equal focus to long-tailed categories, favoring top-1 accuracy. However, CSOT may mislabel some major class samples into unrelated categories, hampering top-5 accuracy. Nevertheless, we believe our CSOT can be extended to class-imbalanced scenarios in future work.
>
> Secondly, the existing work NCE employs a co-training scheme with two models, which can well avoid memorizing the noise. Conversely, we solely train one model, which may lead to incorrect labels being memorized.
>
>
> > **Q6**. How does the batch size affect the empirical performance? How does the number of total classes in the training set affect the choice of the ideal batch size for the OT step?
>
> **A6**. Thank you for raising the concern. Firstly, we would like to claim that to fully capture the local and global structure of the data, more samples per class is better. However, a larger batch size enlarges the OT matrix, potentially affecting computational efficiency. To balance accuracy and efficiency, we guarantee each class has at least 20 samples and set a batch size of 2000 for offline denoising and relabeling. Additionally, we employ an accumulated memory bank to store each mini-batch sample to fill 2000, which avoids the GPU memory limit exceeded problem.

---

> > ### Comment · Reviewer_7s92 · 2023-08-19
> >
> > Thanks for the authors' reply. I keep my score unchanged.

---

### Official Review · Reviewer_Zh5c · 2023-07-05

**Soundness:** 3 good
**Presentation:** 2 fair
**Contribution:** 3 good
**Rating:** 5
**Confidence:** 3

**Summary:**

This paper introduces a novel formulation of Optimal Transport (OT), named Curriculum and Structure-Aware Optimal Transport, for generating pseudo labels by considering both inter- and intra-distribution structures of samples. Moreover, to efficiently estimate the distribution's structure, the authors adopt a curriculum paradigm to progressively train the proposed denoising and relabeling allocator. Additionally, they present a computation method for the proposed CSOT that ensures faster processing speeds, reducing computational overhead. Experimentally, this paper achieved SOTA performance on various benchmarks.

**Strengths:**

1.Estimating the intra- and inter-structure coherence of samples is a convincing and reliable method for improving relabeling accuracy. The proposed prediction-level and label-level consistency constraints seem also interesting and plausible.
2.The combination of the proposed OT and curriculum learning for solving INL is also smooth and makes sense.

**Weaknesses:**

1.The effectiveness for alignment of global and local structures between samples and classes is not fully convincing. The ablation results for OT in Table 3 seem weak. The comparison between row (a) with 78.07 and row (b) with 78.65 suggests that the performance improvement brought by the proposed prediction-level and label-level constraints is limited. Moreover, the introduction of two additional constraints adds complexity to the optimization. Similarly, row (e), CSOT w/o Ω^{L}, achieves the best performance, indicating that the benefits brought by the prediction-level and label-level constraints are unstable.


2.As the part that readers are most concerned about, the section 4.3, the loss function needs to be able to reflect the integrity of the method and the specific combination with its own innovation points. I can’t see the innovation of this article in the loss function here, and each loss term is an existing work. The work in this paper seems to be only used to build a dataset $\mathcal{D}_{\text{clean}}$ and  $\mathcal{D}_{\text{corrupted}}$ for training? This structure and method of writing can greatly weaken the contribution of this paper. Besides, I would like to suggest the authors provide an overall alghrothim to show the whole training process, where the proposed methods would have been used during each training epoch and the training objectives are not the key point in this paper.

3.Considering the complexity of the proposed algorithm, and its marginal improvement over previous methods on two real datasets in Table 2, especially compared to NCE. The effectiveness of this work is questionable. Additional experiments are suggested, especially on Clothing1M. Besides, some related works[1] should be discussed which are published recently.

 [1] OT-Filter: An Optimal Transport Filter for Learning with Noisy Labels (CVPR 2023)

4.The results of row (g) in Table 3 are not sufficient to show that the performance improvement is brought about by the method in this paper. For example, we need a more detailed ablation study to explain the role of NCE loss and the role of CSOT.

5.The structure of the article is confusing, which weakens the contribution of this article. Secondly, the introduction of some tool concepts is quite abrupt. For example, the proposal of Eq. (3) and (6) gives people a new form of OT formulation and will give specific solutions later. This expectation is affected by the entropy regularization in the following text, whose introduction is abrupt. This results in the final Eq. (16) which does not look significantly different from the original sinkhorn algorithm. It is recommended to introduce sinkhorn from the beginning and to emphasize that there are new constraints based on it.

6.The characters in the figures in the experimental part are too small, which affects reading. At the same time, why are there two figures, and their labels are figure 2? Moreover, Fig. 4 is quoted in the description of the text, but there is no figure 4 in fact.

**Questions:**

1.For more precision, I would like to suggest providing a more detailed description of Equation (6). What does the left constraint in Equation (6) signify? Why should $Q\mathbb{1}_{C}$ not exceed $\frac{1}{B}\mathbb{1}_{B}$? Furthermore, it seems that $\frac{m}{c}\mathbb{1}_{C}$ is not a simplex. Could you please explain the rationale behind this relationship and provide additional descriptions or citations to support it? Additionally, the introduction of curriculum into SOT appears to be too direct and lacks smoothness.

2.The identification of selected samples in Equation (7) is confusing. What does "topK(W,⌊mB⌋)" mean? Based on my understanding, it implies that a sample pair (x_i, y_i) will be considered clean if it belongs to the top k convincing samples in the current training batch. I would suggest adding a description for the round-down symbol ⌊ and ⌋ to clarify its purpose.

**Limitations:**

The authors describe its limitations.

---

> ### Author Rebuttal · Authors · 2023-08-09
>
> We thank the reviewer for the constructive feedback on our paper. Here we address your detailed comments below:
>
> > **Q1(1)**. The comparison between classical OT (row(a)) and Structure-aware OT (row(b)) in Table 3 suggests that the performance improvement brought by the proposed two regularization terms is limited. Row(e), CSOT w/o Ω^{L}, achieves the best performance, indicating that the benefits brought by two regularization terms are unstable.
>
> **A1(1)**. Firstly, **the unsurprising improvement of Structure-aware OT (SOT) results from the inherent characteristics of the Learning with Noisy Labels (LNL) task**, where the feature space and noisy labels exhibit ambiguity in the early training stage, leading to biased label generation.
>
> **To ensure reliable pseudo-labels for LNL task, we introduce curriculum constraints to SOT, i.e. our CSOT, where structure-aware regularization terms and curriculum constraints mutually enhance each other.**
>
> As shown in Table3, our CSOT significantly boosts the classical OT (row(a)) performance from 78.07 to 81.85, achieving the best result.
>
> As shown in Figure S6 (Appendix B.5), the CSOT design ensures the high accuracy of clean labels during the early training stage, which helps CSOT build a robust allocator in a high noise rate scenario.
>
>
> > **Q1(2)**. The introduction of two additional constraints adds complexity to the optimization.
>
> **A1(2)**. Our proposed efficient solver ensures minimal computational burden. Table 5 demonstrates our method's faster total training time compared to DivideMix and NCE.
>
>
> > **Q2(1)**. The loss function needs to be able to reflect the integrity of the method and the specific combination with its own innovation points.
>
> **A2(1)**. Firstly, we respectively disagree with this comment. We believe the loss function is not a necessary part to reflect the integrity and novelty of the method. In this paper, we do design a new objective function for OT, i.e. CSOT, incorporating of local coherent regularization terms and curriculum constraints. **Our proposed CSOT and the new solver instead of the loss function are the key innonvation of this paper**. Notably, our CSOT, aiming at identifying clean labels and correcting corrupted labels, is easy to be compatible with other Learning with Noisy Labels (LNL) losses.
>
> Secondly, we would like to claim that **modeling the sample selection and label correction [37][53] are also common directions for LNL as shown in the [R1], which clearly categorizes recent methods into four directions including "Sample Selection" and "Robust Loss Design"**.
>
> [R1] [TNNLS 2022] Learning from noisy labels with deep neural networks: A survey
>
>
> > **Q2(2)**. Suggest the authors provide an overall algorithm to show the whole training process.
>
> **A2(2)**. We provided the overall algorithm in Algorithm S3 (Appendix A.3).
>
>
> > **Q3(1)**. Additional experiments on Clothing1M are suggested.
>
> **A3(1)**. We conducted the experiment on the Clothing1M dataset in Table S7 (Appendix B.5).
>
>
> > **Q3(2)**. The related work [CVPR 2023] OT-Filter should be discussed.
>
> **A3(2)**. Thanks for your advice. We discuss this work in the following paragraph, and they will be involved in our final version.
>
> Firstly, we clarify that OT-Filter is a concurrent study, publicly accessible post the NeurIPS 2023 submission deadline.
>
> Secondly, our proposed CSOT methodology differs significantly from OT-Filter. OT-Filter involves standard OT-based pseudo-labeling with extra sparsity regularization, whereas CSOT concurrently considers both inter- and intra-distribution structures of samples to construct a robust curriculum allocator for denoising and relabeling.
>
> Thirdly, CSOT's superiority over OT-Filter on CIFAR-10/100 and Clothing 1M dataset is demonstrated in Table G1, G2, G3 (global response pdf).
>
>
> > **Q4**. Need a more detailed ablation study to explain the role of NCE loss and the role of CSOT.
>
> **A4**. Firstly, **we did conduct detailed ablation study in Figure 2 (page 9)** with NCE and CSOT, comparing the performance of clean label identification and corrupted label correction. We will enlarge the font size in Figure 2 to make it more clear. **Despite using a similar learning objective as NCE, CSOT excels in sieving and relabeling, confirming our CSOT's effectiveness.**
>
> Secondly, our CSOT serving as a sample selection and pseudo-labeling step, akin to NCE. Thus, we adopt NCE's existing semi-supervised learning objectives, which is not a specially tailored LNL loss.
>
>
> > **Q5**. It is recommended to introduce sinkhorn from the beginning and to emphasize that there are new constraints based on it.
>
> **A5**. Thanks for your advice. We will move Line 196-200 (Section 5) to Section 3 Preliminaries to introduce Sinkhorn algorithm from the beginning in our final version.
>
>
> > **Q6**. Figure readability including mislabeled index and font size.
>
> **A6**. Thank you for advice. Figure 2 on page 9 is wrongly identified as Figure 4. We will revise these in our final version.
>
>
> > **Q7**. More detailed description of the curriculum constraint in Eq.(6). Why should $Q 1_C$ not exceed $\frac{1}{B} 1_B$? Furthermore, it seems that $\frac{m}{C} 1_{C}$ is not a simplex.
>
> **A7**. Firstly, the equality constraint $\alpha=\frac{1}{B}1_B$ in Eq.(2) signifies equal assignment budget for each sample, demanding uniform mapping intensity to the class centroid. **For the purpose of partial assignment, our CSOT relaxing this equality as $Q1_C\leq\frac{1}{B}1_B$.**
>
> Secondly, **while maintaining column equality ($Q^T1_B=\frac{m}{C}1_C$), $m\in[0,1]$ regulates total coupling sum to control the curriculum budget**, as in Line 123-124. We visually demonstrate CSOT's coupling matrix in Figure S4 (Appendix B.2), revealing the curriculum factor $m$ effect.
>
>
> > **Q8**. Suggest adding a description for the symbol ⌊·⌋ to clarify top-k selection in Eq.(7).
>
> **A8**. Thanks for your advice. We will revise this in our final version.

---

> > ### Comment · Reviewer_Zh5c · 2023-08-22
> > **Response to Rebuttal**
> >
> > Thanks for the authors' reply. I increase my score to 5.

---

> ### Author Response · Authors · 2023-08-18
>
> Thanks for your valuable comments and suggestions. We greatly appreciate your time and effort in reviewing our work. We have carefully considered each of your concerns and have made the necessary revisions to address them.
>
> We sincerely hope that our responses have adequately addressed the concerns raised in your review. Please feel free to let us know if you have any further questions. We are dedicated to further clarifying and addressing any remaining issues to the best of our ability.

---

### Official Review · Reviewer_ZdDn · 2023-07-05

**Soundness:** 3 good
**Presentation:** 2 fair
**Contribution:** 3 good
**Rating:** 6
**Confidence:** 4

**Summary:**

This paper introduces CSOT, an approach to address the challenge of noisy labels in machine learning models. CSOT incorporates optimal transport formulation to assign reliable labels during training, considering the structure of the sample distribution. The authors also propose an efficient computational method for solving CSOT. Experimental results demonstrate the superior performance of CSOT compared to existing methods for learning with noisy labels.

**Strengths:**

- The paper is strongly motivated by theoretical analysis, particularly optimal transport analysis.
- The writing style is clear and easy to follow.
- CSOT exhibits superior performance when compared to previous algorithms.

**Weaknesses:**

- The paper lacks a comparison with a baseline algorithm called UNICON [1], which has shown good performance in highly noisy scenarios (e.g., 0.9 noisy ratio). It would be valuable for the authors to include a performance comparison with UNICON.
- The authors do not analyze the case of instance-wise noisy labels, which is a prevalent type of noisy label model. Including an analysis of this case would be beneficial.
- The paper does not investigate the sensitivity of hyperparameters, which are required to run the algorithm. It would be valuable for the authors to perform a hyperparameter sensitivity analysis.
- To enable a comprehensive comparison, the authors should report both the best and last performances of the model, as models trained on noisy labels tend to memorize the noisy labels.

[1] UNICON: Combating Label Noise Through Uniform Selection and Contrastive Learning

Minor)
The legend size in Figure 2 is too small to read.

**Questions:**

-

**Limitations:**

The limitations of this paper are summarized in the "Question" and "Weakness" sections
------
(Raise score from 5 to 6 after rebuttal)

---

> ### Author Rebuttal · Authors · 2023-08-09
>
> We thank the reviewer for the constructive comments which helped us improve the quality of our work. In the following, we have provided a point-by-point response to the comments.
>
> > **Q1**. Lack of a comparison with a baseline algorithm called UNICON.
>
> **A1**. Thanks for your constructive suggestion. We will add UNICON as one of the baseline methods in our final version. And here is a simplified comparison table for UNICON and our CSOT in CIFAR-10/100 dataset, which shows that **our CSOT still outperforms UNICON on CIFAR 100 under high noise ratios**.
>
> Table 1. Comparison between UNICON and CSOT in test accuracy (\%) on CIFAR-10.
> |        |sym-20\%|sym-50\%|sym-80\%|sym-90\%|asym-40\%|
> |  ----  | ----  | ----  | ----  | ----  |  ----  |
> |UNICON| 96.0 | 95.6 | 93.9 | **90.8** | 94.1  |
> |CSOT (Ours)    | **96.6** | **96.2** | **94.4** | 90.7 |  **95.5**  |
>
> Table 2. Comparison between UNICON and CSOT in test accuracy (\%) on CIFAR-100.
> |        |sym-20\%|sym-50\%|sym-80\%|sym-90\%|
> |  ----  | ----  | ----  | ----  | ----  |
> |UNICON| 78.9 | 77.6 | 63.9 | 44.8 |
> |CSOT (Ours)    | **80.5** | **77.9** | **67.8** | **50.5** |
>
>
>
> > **Q2**. Supplement an analysis of the case of instance-wise noisy labels.
>
> **A2**. Thanks for your constructive suggestion. We validate the effectiveness of our proposed method in instance-wise noise under the noise rate of 20\%, 40\%, 60\% in the following table.
>
> Table 3. Comparison of instance-wise noise among DivideMix, NCE, and CSOT in test accuracy (\%) on CIFAR-10.
> |  noise rate  |20\%|40\%|60\%|
> |  ----  | ----  | ----  | ----  |
> |DivideMix| 92.26 | 93.86 | 54.34 |
> |NCE| 96.00 | 95.24 | 75.58 |
> |CSOT (Ours)    | **96.21** | **95.82** | **76.65** |
>
> Table 4. Comparison of instance-wise noise among DivideMix, NCE, and CSOT in test accuracy (\%) on CIFAR-100.
> |   noise rate |20\%|40\%|60\%|
> |  ----  | ----  | ----  | ----  |
> |DivideMix| 77.52 | 73.58 | 38.72 |
> |NCE| **80.40** | 74.92 | 70.34 |
> |CSOT (Ours)    | 79.22 | **76.55** | **72.71** |
>
>
> > **Q3**. Supplement the sensitivity of hyperparameters.
>
> **A3**. We analyzed the sensitivity of hyperparameters $\varepsilon$ and $\kappa$ in our CSOT in supplementary materials, please refer to Figure S7 and Figure S8 (Appendix B.6).
>
>
> > **Q4**. Supplement both the best and last performances of the model.
>
> **A4**. We have reported both the best and last performances for CIFAR-10/100 dataset in Table 1. Following the existing method, we only report the last performance on the WebVision and Clothing1M dataset.
>
>
> > **Q5**. The legend size in Figure 2 is too small to read.
>
> **A5**. Thank you for pointing this out. We will adjust it to a more readable size in our final version.

---

> > ### Comment · Reviewer_ZdDn · 2023-08-17
> > **Official Comments and decision.**
> >
> > Thank you for your detail responses.  I have read the responses from the authors to the questions I raised (including other reviewers' comments and responses). I would like to raise the score from 5 to 6.

---

### Official Review · Reviewer_gpRK · 2023-07-05

**Soundness:** 4 excellent
**Presentation:** 3 good
**Contribution:** 4 excellent
**Rating:** 7
**Confidence:** 5

**Summary:**

This paper proposes a novel optimal transport formulation, called Curriculum and Structure-aware Optimal Transport (CSOT) for learning with noisy labels. CSOT considers both the inter- and intra-distribution structure of the samples to construct a robust denoising and relabeling allocator. It’s worthing noting that Notably, CSOT is a new OT formulation with a nonconvex objective function and curriculum constraints. The authors developed a lightspeed computational method that involves a scaling iteration within a generalized conditional gradient framework to solve CSOT efficiently.

**Strengths:**

1. Novel Approach: The paper proposes a novel optimal transport (OT) formulation called Curriculum and Structure-aware Optimal Transport (CSOT) to address the challenge of learning with noisy labels. I believe this paper introduces a new perspective and potentially brings fresh insights to the field.

2. Consideration of Global and Local Structure: Unlike current approaches that evaluate each sample independently, CSOT concurrently considers the inter- and intra-distribution structure of the samples. This consideration of both global and local structure helps construct a more robust denoising and relabeling allocator, potentially leading to improved performance.

3. Incremental Assignment of Reliable Labels: CSOT incrementally assigns reliable labels to a fraction of the samples with the highest confidence during the training process. This approach ensures that the assigned labels have both global discriminability and local coherence, which could contribute to better generalization and reduced overfitting.

4. This paper provides a very detailed derivation for the lightspeed computational method.

**Weaknesses:**

Researchers or practitioners interested in using CSOT may need to invest additional effort in adapting or developing specialized solvers.

**Questions:**

1. How does CSOT incorporate the global and local structure of the sample distribution? Could you provide more details on the methodology used to capture this information?

2. Can you explain in more detail the nonconvex objective function of CSOT? How does the nonconvexity affect the optimization process, and how does the proposed method handle this challenge?

3. What is the iteration number required by the generalized conditional gradient?

4. Are there any specific scenarios or types of noisy labels where CSOT may not perform as well?


**Limitations:**

CSOT is described as having a nonconvex objective function. Nonconvex optimization problems can be challenging to solve, and they may have multiple local optima, making it difficult to guarantee finding the global optimum. This could potentially impact the reliability and efficiency of the proposed method.

---

> ### Author Rebuttal · Authors · 2023-08-09
>
> We thank the reviewer for our paper's positive feedback and constructive suggestions. Here are our responses to the reviewer's comments.
>
> > **Q1**. Researchers or practitioners interested in using CSOT may need to invest additional effort in adapting or developing specialized solvers.
>
> **A1**. Thank you for raising the concern.
>
> Firstly, We would like to justify that **our proposed solver is available for arbitrary differentiable regularization term $\Omega$, marginal constraints vectors $\alpha$, and $\beta$** in Eq.(16), as we stated in Algorithm 2 Input row (Line 1).
>
> Secondly, **our code provides a versatile solver** (Algorithm 2) API that can be used with any user-defined $\Omega$, $\nabla\Omega$, $\alpha$, and $\beta$. This feature simplifies the process for follow-up researchers to conduct their experiments, especially for those who wish to validate the effectiveness of their own customized regularization terms $\Omega$. As a result, researchers do not have to spend time adapting or developing specialized solvers, thus enhancing the efficiency of their work.
>
> Thirdly, we highlight that **our work provides a valuable example of developing a customized OT formulation and a corresponding solver**. By showcasing our method's adaptability and efficacy, we aim to contribute to the wider application and exploration of OT across various domains.
>
>
> > **Q2**. Provide more details about incorporating the global and local structure of the sample distribution in CSOT.
>
> **A2**. Firstly, the local structure, i.e. intra-distribution coherence among samples, is preserved by two local coherent regularized terms defined in Equations (4) and (5). Technically speaking, to formulate the coherent regularized terms, we construct the correlation among $i$-th sample, $j$-th sample, and $k$-th class centroid by element-wise multiplication. As shown in Figure 1 Top, classical OT tends to mismatch two nearby samples to two far-away class centroids when the decision boundary is not accurate enough. To mitigate this, our SOT generates local consensus assignments for each sample by preserving prediction-level and label-level consistency. Notably, for vague samples located near the ambiguous decision boundary, SOT rectifies their assignments based on the neighborhood majority consistency.
>
> Secondly, the global structure, i.e. inter-distribution discriminability between samples and categories, is preserved by curriculum constraints $\Pi^c(\alpha,\beta)$ defined in Eq.(11). This property is inherited from the OT constraints $\Pi(\alpha,\beta)$ defined in Section 3 (Line 122), which enforces the marginal distributions of the coupling matrix equal to given samples and categories distribution $\alpha$ and $\beta$. As shown in Figure S3 (Appendix B.1), OT-based Pseudo-Labeling (PL) tends to generate more discriminative labels than prediction-based PL. To explain this, OT-based PL optimizes the mapping problem by considering the inter-distribution matching of samples and categories, rather than the prediction-based PL assigning labels solely in a per-class manner.
>
> Thirdly, our proposed curriculum constraints incorporate the global and local structure of the sample distribution, prioritizing samples with better global discriminability and local coherence properties for label assignment, thereby enabling a robust curriculum allocator.
>
>
> > **Q3**. How does the nonconvexity affect the optimization process, and how does the proposed method handle this challenge?
>
> **A3**. The nonconvexity would bring some problems including local minima, rather slow convergence speed than convex case. However, solving nonconvex objectives with the generalized conditional gradient (GCG) algorithm has been supported by strong convergence analysis [R1][R2]. And we find our CSOT converges fast as shown in Figure S5 (Appendix B.4).
>
> Moreover, some OT-like nonconvex problems, such as Gromov-Wasserstein problem, are also solved by GCG algorithm and provide convergence analysis [10][19].
>
> [R1] Bredies K, Lorenz D, Maass P. Equivalence of a generalized conditional gradient method and the method of surrogate functionals[M]. Bremen, Germany: Zentrum für Technomathematik, 2005.
> [R2] Beck A. First-order methods in optimization[M]. Society for Industrial and Applied Mathematics, 2017.
>
>
>
> > **Q4**. Specify the iteration number required by the generalized conditional gradient.
>
> **A4**. As we specified in Section 6.1 (Line 238-239), the number of outer loops is set to 10, and the number for inner scaling iteration is set to 100.
>
>
>
> > **Q5**. Specify the potential scenarios or types of noisy labels where CSOT may not perform as well.
>
> **A5**. As we stated in Section 7, extreme class-imbalance cases are not considered in this paper. In highly imbalanced scenario, the generated pseudo-labels would be biased due to uniformed class distribution vector $\beta$. However, we believe that our approach can be further extended for this purpose in the future work.

---

### Official Review · Reviewer_qnim · 2023-07-12

**Soundness:** 3 good
**Presentation:** 3 good
**Contribution:** 3 good
**Rating:** 5
**Confidence:** 3

**Summary:**

This paper proposes a new noisy label learning approach based on Optimal Transport (OT) and Pseudo-Labeling (PL). Specifically, the authors extent OT-based PL with the consideration of the intrinsic coherence structure of sample distribution. Consequently, this paper proposes a novel optimal transport formulation, namely Curriculum and Structure-aware Optimal Transport (CSOT), which constructs a robust denoising and relabeling allocator that mitigates error accumulation. Experiments on both controlled and real noisy label datasets show the effectiveness of the proposed method.

**Strengths:**

1. The paper proposes a method named Curriculum and Structure-aware Optimal Transport (CSOT) to address the problem of noisy label learning, and the application of OT-based pseudo-labeling in tackling noisy label learning problem has not been thoroughly investigated.

2. The experimental results on different datasets in this paper validate the effectiveness of the proposed method. Additionally, several ablation experiments are conducted to demonstrate the effectiveness of each module in the method.

**Weaknesses:**

1. In terms of methodological novelty, OT-based PL has been previously applied to other problems, and this paper only applies it to the specific problem namely noisy label learning rather than introducing it for the first time. Additionally, employing curriculum learning to address the issue of pseudo-labeling is a common strategy in the field of weakly supervised learning.

2. The utilization of SOT is a key contribution of this paper. However, the current motivation behind this aspect, as presented in Figure 1, is not sufficiently clear. The authors are encouraged to provide additional descriptions in this section to enhance the clarity and understanding of the motivation.

3. Since the differences between the proposed method and the comparison methods in several cases is too small, it is hard to provide a clear comparison without the present of standard deviations. Additionally, the backbone and other parameter settings of the SOTA methods are not clearly listed, thus, further evidence is needed to establish the fairness of the comparisons.


**Questions:**

Please refer to Weaknesses

**Limitations:**

N/A.

---

> ### Author Rebuttal · Authors · 2023-08-09
>
> We thank the reviewer for the constructive comments which helped us improve the quality of our work. In the following, we have provided a point-by-point response to the comments.
>
>
> > **Q1**. OT-based Pseudo-Labeling (PL) is not proposed for the first time and curriculum learning to address the issue of pseudo-labeling is a common strategy.
>
> **A1**. Firstly, we emphasize that directly applying the original OT to certain PL tasks may result in sub-optimal performance. Consequently, it **becomes necessary for researchers to investigate more adaptive OT formulations for specific problems** [9][54], including off-the-shelf variants like unbalanced OT and partial OT, to address the specific challenges of the problem. To this end, we propose a novel CSOT formulation tailored for the denoising and relabeling task. Notably, our CSOT formulation innovatively incorporates two local coherent regularization terms and curriculum constraints, enabling the incremental generation of reliable pseudo-labels for the Learning with Noisy Labels task.
>
> Secondly, we would like to claim that **our work is the first to propose a curriculum scheme that fully considers the inter- and intra-distribution structure of the samples based on OT**, to the best of our knowledge. Also, introducing curriculum scheme to OT requires a new solver and **we innovatively propose a lightspeed computational method**. This approach stands in stark contrast to existing curriculum-based PL methods, setting our work apart in terms of its novelty and uniqueness.
>
> Thirdly, we highlight that our work **provides a valuable example of developing a customized OT formulation and a corresponding solver**. By showcasing our method's adaptability and efficacy, we aim to contribute to the wider application and exploration of OT across various domains.
>
>
> > **Q2**. Provide additional descriptions about the Structure-aware OT in Figure.
>
> **A2**. Thanks for your constructive suggestion. We will add the following descriptions to Figure 1 in our final version.
> "(Top) Comparison between classical OT and our proposed Structure-aware OT. Classical OT tends to mismatch two nearby samples to two far-away class centroids when the decision boundary is not accurate enough. To mitigate this, our SOT generates local consensus assignments for each sample by preserving prediction-level and label-level consistency. Notably, for vague samples located near the ambiguous decision boundary, SOT rectifies their assignments based on the neighborhood majority consistency."
>
> Moreover, to further show the differences among prediction-, OT-, and Structure-aware OT-Based PL, we provided a more intuitive illustration in Figure S3 (Appendix B.1).
>
>
>
>
>
> > **Q3**. Lack of standard deviations and experimental settings of the SOTA methods.
>
> **A3**. Thank you for raising the concern. For the standard deviations, we did provide the results of standard deviations on CIFAR10/100 in Table 1. Here we provide standard deviations for the Webvision dataset as follows:
>
> Table 1. Comparison between NCE and CSOT in top-1/5 test accuracy (\%) on the
> Webvision and ImageNet ILSVRC12 validation sets.
> |        |Webvision-top-1|Webvision-top-5|ILSVRC12-top-1|ILSVRC12-top-5|
> |  ----  | ----  | ----  | ----  | ----  |
> |CSOT (Ours)| 79.67±0.14 | 91.95±0.21 | 76.64±0.16 | 91.67±0.18 |
>
> For the experimental settings, we inherit the same backbone, optimizer parameters from the SOTA works NCE and DivideMix. More details are provided in Appendix A.1, please refer to our supplementary materials.

---

> > ### Comment · Reviewer_qnim · 2023-08-22
> > **Thanks for the responses**
> >
> > Thank you for your responses. Your explanation has clarified some of the previous concerns. However, I still believe that this paper offers limited contributions to the current field of learning with label noise. Therefore, I will maintain my current score.

---

### Author Rebuttal · Authors · 2023-08-09

We thank the reviewers for their careful reading of our paper and help with improving our manuscript. We sincerely appreciate that you find our work:
* proposes novel objective which is a solid idea (Reviewer 7s92)
* proposes a convincing, reliable, interesting and plausible method (Reviewer Zh5c)
* is strongly motivated by theoretical analysis (Reviewer ZdDn)
* provides a very detailed derivation for the lightspeed computational method (Reviewer gpRK)
* exhibits superior and strong performance (Reviewer ZdDn, Reviewer 7s92)
* introduces a new perspective and potentially brings fresh insights to the field (Reviewer gpRK)

We thank the reviewers for their careful reading of our paper and help with improving our manuscript. We sincerely appreciate that you find our work:
* proposes novel objective which is a solid idea (Reviewer 7s92)
* proposes a convincing, reliable, interesting and plausible method (Reviewer Zh5c)
* is strongly motivated by theoretical analysis (Reviewer ZdDn)
* provides a very detailed derivation for the lightspeed computational method (Reviewer gpRK)
* exhibits superior and strong performance (Reviewer ZdDn, Reviewer 7s92)
* introduces a new perspective and potentially brings fresh insights to the field (Reviewer gpRK)

In the subsequent sections, we aim to address the concerns and questions you raised, offering a comprehensive item-by-item response to each of your comments.

We have provided some additional experiments results as reviewers suggest. Due to space limitations, we've displayed the results table **in the global response PDF** for Reviewer Zh5c Q3(2) and Reviewer 7s92 Q2.

---

### Decision · Program_Chairs · 2023-09-21

**Decision:**

Accept (poster)

**Comment:**

The reviewers found the paper interesting, with an OT approach for classification in the presence of label noise and good performances but had a few concerns regarding limited novelty of applying OT for PL and label, limited performance gains, missing references and baselines,  lack of sensitivity analysis, which lead to a borderline positive score.

The authors did a good rebuttal by providing new numerical experiments and clarification to the reviewers concerns. the rebuttal was appreciated and two of the most negative reviewers increased their score. The final decision is to accept the paper but strongly suggest that the authors take into account all the reviewers comments in the main paper and supplementary including discussing the missing references and implementing them (all the discussed OT for noise label methods) and providing variances on more than 3 realizations in the final paper. An effort should also be made to explain better the proposed regularization, in particular the relation of equation (3) with quadratic OT formulations (Fused Gromov-Wasserstein, Laplacian regularized OT (Ferradans 2014) ) whose optimizer was used.

List of methods discussed by reviewers that should be discussed and compared (OT-Filter is contemporary but the authors can be commended since they compared to it in the rebuttal):

Karim, N., Rizve, M. N., Rahnavard, N., Mian, A., & Shah, M., 2022, Unicon: Combating label noise through uniform selection and contrastive learning. In Proceedings of the IEEE/CVF Conference on Computer Vision and Pattern Recognition (pp. 9676-9686).

Feng, C., Ren, Y., & Xie, X. , 2023, OT-Filter: An Optimal Transport Filter for Learning With Noisy Labels. In Proceedings of the IEEE/CVF Conference on Computer Vision and Pattern Recognition (pp. 16164-16174).

Damodaran, B.B., Flamary, R., Seguy, V. and Courty, N., 2020. An entropic optimal transport loss for learning deep neural networks under label noise in remote sensing images. Computer Vision and Image Understanding, 191, p.102863.